# psiCLIP reveals dynamic RNA binding by DEAH-box helicases before and after exon ligation

Lisa M. Strittmatter [1,6], Charlotte Capitanchik[2,6], Andrew J. Newman[1], Martina Hallegger[2,3], Christine M. Norman[1], Sebastian M. Fica [1,7], Chris Oubridge[1], Nicholas M. Luscombe [2,4,5,7], Jernej Ule [2,3,7 ✉] & Kiyoshi Nagai [1]

RNA helicases remodel the spliceosome to enable pre-mRNA splicing, but their binding and mechanism of action remain poorly understood. To define helicase-RNA contacts in specific spliceosomal states, we develop purified spliceosome iCLIP (psiCLIP), which reveals dynamic helicase-RNA contacts during splicing catalysis. The helicase Prp16 binds along the entire available single-stranded RNA region between the branchpoint and 3'-splice site, while Prp22 binds diffusely downstream of the branchpoint before exon ligation, but then switches to more narrow binding in the downstream exon after exon ligation, arguing against a mechanism of processive translocation. Depletion of the exon-ligation factor Prp18 destabilizes Prp22 binding to the pre-mRNA, suggesting that proofreading by Prp22 may sense the stability of the spliceosome during exon ligation. Thus, psiCLIP complements structural studies by providing key insights into the binding and proofreading activity of spliceosomal RNA helicases.

[1] MRC Laboratory of Molecular Biology, Cambridge, UK. [2] The Francis Crick Institute, London, UK. [3] Department of Neuromuscular Diseases, UCL Queen Square Institute of Neurology, Queen Square, London, UK. [4] UCL Genetics Institute, Department of Genetics, Environment and Evolution, University College London, London, UK. [5] Okinawa Institute of Science & Technology Graduate University, Okinawa, Japan. [6]These authors contributed equally: Lisa M. Strittmatter, Charlotte Capitanchik. [7]These authors jointly supervised this work: Sebastian M. Fica, Nicholas M. Luscombe, Jernej Ule. ✉email: jernej.ule@crick.ac.uk

Splicing is an essential step in pre-mRNA processing in eukaryotes, during which non-coding introns are removed and coding exons are ligated. Splicing is catalysed by a large, dynamic molecular machine called the spliceosome, which comprises about a hundred proteins in yeast. During the first splicing step—branching—the spliceosome catalyses attack by the branch point adenosine (brA) at the 5′-splice site (5′-SS), producing a lariat-intermediate structure[1]. During the second step of splicing—exon ligation—the cleaved 5′-SS attacks the 3′-splice site (3′-SS) to ligate the exons and form the mature RNA (mRNA). To faithfully remove introns, the spliceosome must accurately recognize the 5′-SS, the brA, and the 3′-SS (Fig. 1A). Use of the wrong splice sites could ultimately lead to errors in protein translation and erroneous splicing has been linked to cancer in humans, leading to the development of therapeutics designed to modulate splicing[2]. Eight ATP-dependent helicases—comprising three DEAD-box helicases (Sub2, Prp5, and Prp28), four DEAH-box helicases (Prp2, Prp16, Prp22, and Prp43), and a Ski2-like helicase (Brr2)—ensure splicing fidelity by actively promoting correct spliceosome assembly and splice site usage[3]. All eight helicases are essential in yeast and conserved in humans[4,5].

The DEAD-box helicases Sub2, Prp5, and Prp28 are involved in early assembly of the spliceosome on introns, promoting correct splice site recognition[5]. The Ski2-like helicase Brr2 is involved in spliceosome activation prior to the first catalytic step of splicing. The four DEAH-box helicases Prp2, Prp16, Prp22, and Prp43 remodel the spliceosome through the first (branching) and second (exon ligation) catalytic steps, and coordinate spliceosome disassembly once the reaction has completed[5]. In vitro experiments and structural studies of spliceosomal DEAH-box helicases showed that they translocate on RNA in the 3′ to 5′ direction to unwind duplexes[6,7]. Prp16 repositions the substrate after the branching reaction by removing the branch helix, formed between the branch point sequence and U2 snRNA, from the active site of the C-complex spliceosome[8–10]. Following this remodelling event, Prp16 dissociates from the C complex and causes dissociation of the branching factors that stabilize the branch helix[11]. Prp16 action generates binding sites for the exon-ligation factors Slu7 and Prp18 (refs. [10,12,13]), resulting in formation of the C* complex, which is competent for exon ligation. In the C* complex, Prp16 is replaced by Prp22, which proofreads exon ligation[14]. Together with the exon-ligation factors Slu7 and Prp18, Prp22 promotes exon ligation for most pre-mRNA substrates, acting in an ATP-independent manner[15]. Exon ligation results in the formation of the spliceosomal P complex that retains both the excised intron and the mature mRNA, which is held by spliceosomal proteins and the U5 snRNA[11]. The mature mRNA is then released from the spliceosome through the ATP-dependent activity of Prp22 (refs. [15–17]), which also results in the release of the exon-ligation factors Prp18 and Slu7. Prp22 was shown to unwind duplex RNA in vitro in an NTP-dependent manner[6,17]. After exon ligation, Prp22 crosslinks to the second exon and was proposed to release the mRNA by translocating along the second exon in a 3′ to 5′ direction, thus disrupting the U5 snRNA–mRNA duplex[18]. Nonetheless, it remained unclear whether Prp22 translocates directly through the duplex or acts at a distance.

Despite advances in structural and compositional analysis of the spliceosome, much is still unknown about helicase-mediated transitions between spliceosome intermediates[11,19]. Although previous research indicates the stages at which helicases are required, it is not known exactly where on the snRNA and/or pre-mRNA they bind, how they promote conformational changes in the spliceosome, nor the mechanism by which they ensure splicing fidelity. Both steps of splicing are reversible, offering the

potential to prevent suboptimal pre-mRNA transcripts from completing splicing[20]. Indeed, DEAD/H-box helicases were proposed to promote the backwards reaction for suboptimal splice sites that engage the spliceosome active site[3,21]. Two helicase-mediated proofreading mechanisms were suggested to increase splicing fidelity. The kinetic model posits that the ATPase activity of DEAH-box helicases acts as a timer. ATP hydrolysis promotes either the productive pathway towards exon ligation, for optimal splice sites, or the discard pathway, for suboptimal splice sites, depending on the rate of catalysis, which is inferred to be faster than ATP hydrolysis for optimal substrates[3]. In the discard pathway, spliceosomes assembled on suboptimal splice sites are dissociated by the ATPase Prp43, whose activity safeguards against the accumulation of splicing errors[22]. The thermodynamic model proposes that splicing fidelity is enhanced through modulating the stability of competing spliceosome conformations, with suboptimal substrates being rejected because they promote conformations that compete with catalysis[3]. Indeed, DEAH-box helicases were proposed to shift the equilibrium between different catalytic conformations of the spliceosome[23]. Although these two models are not mutually exclusive, their relative role in proofreading is unknown.

Most helicases bind to the spliceosome only transiently, making them difficult to study. Electron cryo-microscopy (cryo-EM) structures have shown helicases to be at the spliceosome periphery, where they are detected at much lower local resolutions than other components due to their flexibility. Given their peripheral location in these structures, it remains unclear by what mechanisms helicases promote remodelling of the RNA-based catalytic core of the spliceosome.

Additional insights have been gained by site-specific cross-linking of helicases to pre-mRNA substrates in splicing extracts upon incorporation of single photo-reactive residues into the RNA[18,24,25]. However, the identified binding positions were limited to the nucleotides chosen for site-specific modification and were not linked to a specific conformation of the spliceosome. Single-molecule fluorescence resonance energy transfer (smFRET) studies have advanced our understanding of the mechanism of DEAH-box helicases, suggesting they remodel the spliceosome active site from a distance[9]. However, this approach does not provide precise positional information about RNA–protein contacts and is technically cumbersome, as the positioning of FRET probes must be carefully calibrated for each substrate. Perhaps for this reason, most studies focused on one specific pre-mRNA transcript, making it difficult to distinguish transcript-specific effects from general principles.

Transcriptome-wide studies, such as spliceosome iCLIP using an antibody against the small nuclear ribonucleoprotein-associated protein B (SmB)[26], profiling using Prp19-TAP affinity purification[27], and affinity purification of post-catalytic spliceosomes after mRNA release[28], have uncovered new positional principles for assembly of specific spliceosomal complexes on pre-mRNAs. However, the contribution of individual splicing factors is difficult to discern in such profiles. In addition, iCLIP of individual splicing factors has revealed their position-dependent capacity to control alternative splicing decisions in mammals[29], but has not led to direct insights into step-specific mechanisms for the core spliceosome. A key issue with these methods is that spliceosomes are not stalled at a specific step, therefore the resulting profiles represent an ensemble of spliceosomal snapshots across many splicing steps, with a likely bias towards those that are rate limiting. This prevents us from assigning specific binding sites to individual steps. To understand the molecular mechanisms that underlie splicing fidelity and splice site choice, a method is needed that is capable of identifying RNA contacts formed at defined steps of splicing. Such a method would be

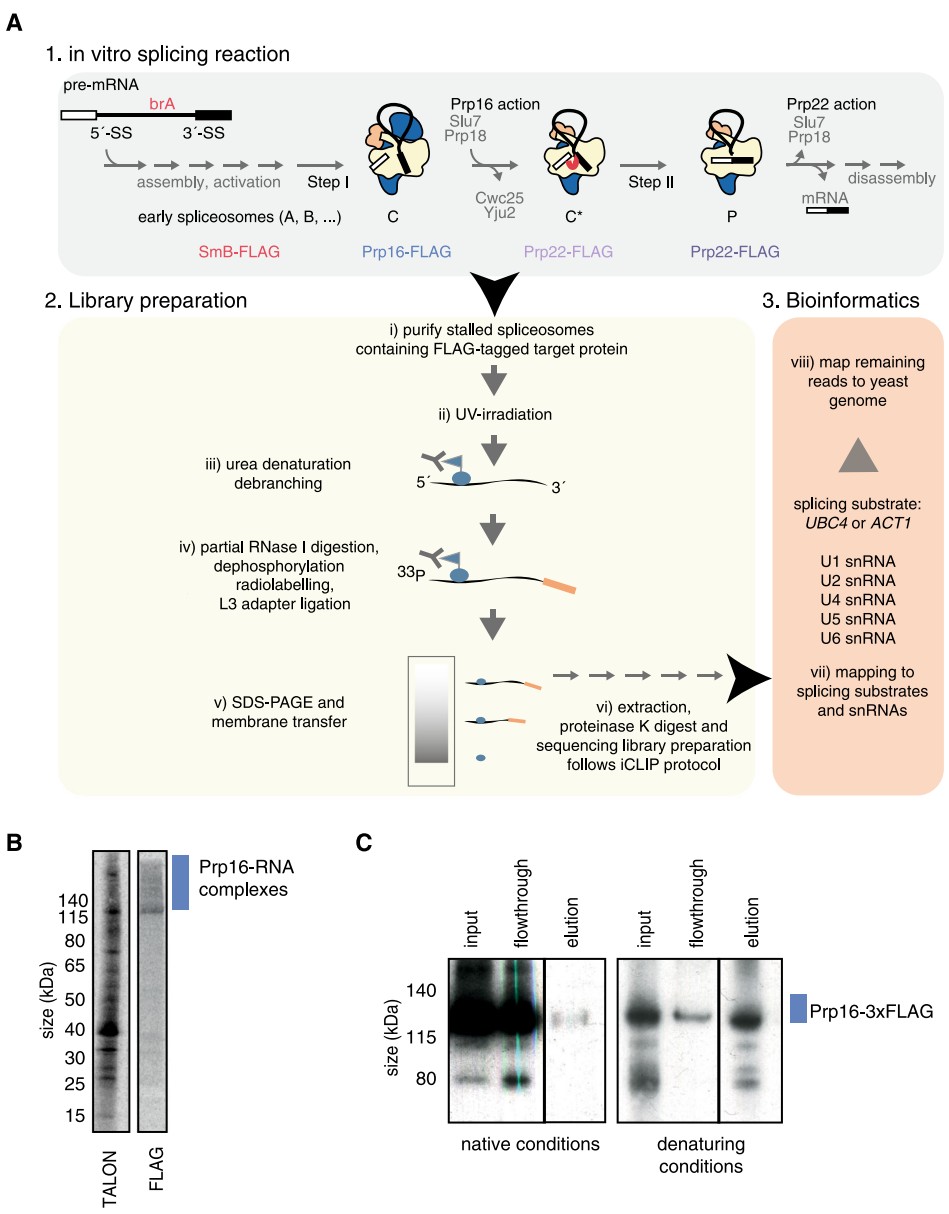

**Fig. 1 The psiCLIP method reveals the complete RNA-binding profile of an RBP in a specific spliceosomal complex. A** Spliceosomes are assembled from yeast cell extract on an in vitro transcribed pre-mRNA and stalled through different strategies that block the transition towards later stages. Spliceosomes are depicted in schematics, where U2 snRNP is coloured in yellow, U5 snRNP in blue, and U6 snRNP in orange. The RNA-binding protein of interest is fused to a C-terminal FLAG tag. After purification of the stalled spliceosome population, spliceosomes are irradiated with a defined UV dose at 254 nm, and denatured to release the target protein from the complex and allow FLAG capture of the RNA–protein complex. For spliceosomes stalled at steps containing lariat structures, an additional debranching step was introduced. Controlled RNase I digestion fragments the RNA into suitable pieces for Illumina NGS. The 5′ end is radio-labelled, while a DNA adapter is attached to the 3′ end, carrying a sequence for primer annealing for reverse transcription. Finally, the crosslinked RNA–protein complexes are resolved on SDS–PAGE and extracted for RNA library preparation. See "Methods" for details. **B** Autoradiographs of RNA after the psiCLIP procedure following capture of Prp16 with either TALON beads for His-tag proteins (left) or anti-FLAG beads for 3× FLAG tag proteins (right). Signal that arises from Prp16 crosslinked to RNA is expected to be larger than the size of Prp16 at 122 kDa. **C** anti-FLAG western blot detecting Prp16-3× FLAG shows that denaturing conditions are required to capture the tagged protein. Purified spliceosomal complex C containing endogenous Prp16-3× FLAG was captured using anti-FLAG beads after either treatment with 6 M urea (right) or under native conditions (left). In **B** and **C**, a representative autoradiograph for wild-type Prp16 is shown; a similar result was obtained with Prp16-G378A.

particularly useful in resolving the principles that guide the binding of spliceosomal helicases to RNA substrates, in identifying potential transcript-specific effects, and for understanding how the ATPase activities of helicases are linked to their RNA-binding behaviour and thereby to their proofreading activities.

Here, we present purified spliceosome iCLIP (psiCLIP), a method to determine the complete range of contacts between specific splicing substrates and helicases in native spliceosomal complexes stalled at specific stages of the splicing cycle. UV crosslinking of an enriched complex makes it possible to capture the weak and transient binding of DEAH-box helicases. We used a system of in vitro-assembled spliceosomes in budding yeast (*Saccharomyces cerevisiae*), which allows targeted isolation of helicases at defined points in the splicing pathway and enables functional characterization of helicases using specific mutants. We used psiCLIP to analyse two DEAH-box helicases, Prp16 and

Prp22, which act before and after exon ligation, respectively (C, C*, and P complexes), in the context of multiple pre-mRNA transcripts. The high sensitivity and resolution of our psiCLIP data provide insights into helicase dynamics and lay the foundation for future mechanistic studies to dissect helicase functions in the spliceosome.

## Results

**Optimized psiCLIP detects RNA binding with positional specificity.** In addition to conventional iCLIP, which uses crosslinked cells or tissues as starting material[30], iCLIP has also been adapted to study the binding of purified U2AF2 to pre-mRNAs in vitro[31]. Here we establish psiCLIP, which adapts iCLIP for in vitro studies of spliceosomes stalled at defined stages of the splicing cycle. Native spliceosomes from yeast cell extracts are assembled in vitro on a defined pre-mRNA substrate and stalled, using substrate modifications or dominant-negative, recombinant helicases. Enriched step-specific spliceosomes are then purified in a similar way to those used for structural analysis[32] (Fig. 1A), using affinity tags on the specific pre-mRNA substrate (Supplementary Fig. 1A). As in conventional iCLIP, spliceosomes are irradiated with 254 nm ultraviolet light to crosslink proteins to the pre-mRNA substrate and snRNAs. Next, the RNA-binding protein (RBP) of interest is purified under stringent, denaturing conditions, along with any crosslinked RNA, followed by cDNA library preparation and high-throughput sequencing. The computationally processed sequencing reads provide a profile of RBP–RNA contacts that occur within the spliceosome enriched for a specific state (see Methods).

In this study, we used psiCLIP primarily to investigate the binding profiles of the DEAH-box helicases Prp16 and Prp22 that act in the spliceosomal C, C*, and P complexes (Fig. 1A). Therefore, we optimized purification, crosslinking, and RNA fragmentation for these helicases. Purification using a 3× FLAG tag was more specific than His-tag purification via TALON beads, which has been a common choice for crosslinking and immunoprecipitation[33], therefore, we proceeded with 3× FLAG for all experiments in this study (Fig. 1B). We found that stringent 6 M urea denaturing conditions were required to capture 3× FLAG-tagged Prp16 bound to a range of RNA fragment sizes (Fig. 1C). To optimize crosslinking and RNA fragmentation, we titrated UV and RNase I doses in parallel (Supplementary Fig. 1B), which showed that the signal was saturated at a UV dose of $312 \times 100 \, \mu J/cm^2$. We chose the UV dose so as to give efficient crosslinking, while minimizing the potential for multiple proximal protein–RNA crosslinks, which could bias reverse transcription truncations to the 3′ end of the binding sites. We also titrated the amount of RNase for treatment of purified spliceosomes so as to minimize the biases that can result from RNA sequence and structure preferences of the RNase[34,35].

To demonstrate the positional specificity of psiCLIP, we first assayed binding of the integral protein SmB, which is known to bind the Sm site in U1, U2, U4, and U5 snRNAs, and is thus present in spliceosomes at all stages of splicing. The Sm binding sites have been characterized by genetic studies, which identified a U-rich consensus sequence as essential for binding[36], while the Sm proteins have been modelled in crystal and cryo-EM structures[37–40]. However, the exact nucleotide-binding positions of the yeast Sm proteins, including their unstructured tail regions, are unknown for most snRNPs, providing an initial test of our method.

Spliceosomes were assembled on a substrate with a 3′-SS mutation from the canonical 3′-UAG| to 3′-UAC| (where | marks the splice site), which impairs exon ligation. A mixture of pre-catalytic and catalytic (complex C) spliceosomes were purified by glycerol gradient and subsequent substrate affinity purification using MS2 stem–loops on the pre-mRNA substrate. In agreement with a previous study[26], psiCLIP performed under stringent 6 M urea denaturing conditions identified SmB crosslinks primarily on snRNAs, with little signal on the pre-mRNA substrate (Fig. 2A and Supplementary Fig. 2A). We observed crosslinks within, and adjacent to, the expected Sm sites, with little signal in the negative controls (no UV/tagged protein; Fig. 2B, C and Supplementary Fig. 2B–G). The three-dimensional structure for the pre-catalytic spliceosomal complex pre-B[40] shows that the SmB protein is in close spatial proximity to the U1 snRNA nucleotides detected using psiCLIP (Fig. 2D). The 5′ end of U1 snRNA extends into the direction of the SmB protein, thus explaining why psiCLIP detected crosslinks mainly at the beginning of the Sm site motif. In addition, the crosslinks upstream of the Sm site identified by psiCLIP (G535 and A536) interact with the long tail of the SmB protein in the cryo-EM structure. Only part of this SmB tail could be modelled as an alpha-helix into the U1 snRNP in the cryo-EM structure of the yeast pre-B complex, due to its high flexibility[3,6]. The psiCLIP data suggest that SmB has a similar configuration in the other snRNPs (Supplementary Fig. 2B–G). The SmB example shows that psiCLIP detects RNA–RBP interactions with high positional specificity and can also provide information on more transient interactions between flexible protein regions and RNA, which are challenging to detect by structural methods.

**psiCLIP reveals broad binding of Prp16 downstream of brA.** Next, we set out to study the RNA contacts made by spliceosomal helicases. The DEAH-box helicase Prp16 is essential for the transition from complex C, the complex after branching, to complex C*, which catalyses exon ligation[13]. Prp16 binds transiently to the spliceosome to reposition the splicing substrate, removing the brA from the active site, creating space that will later allow the 3′-SS to dock in the active site[10,41]. This Prp16-dependent spliceosome remodelling dissociates branching factors Cwc25 and Yju2, and promotes stable binding of the exon-ligation factors Slu7 and Prp18 (refs. [9,10,12,13,42,43]).

To further optimize psiCLIP, we introduced a debranching step for spliceosomes bound to the lariat RNA structure produced after branching, in order to reduce cDNA truncations at branch points, which could otherwise be erroneously interpreted as protein–RNA crosslinks[26,28] (Supplementary Fig. 3). Despite debranching, we still observed some cDNA truncations at the brA, which likely result from truncation at partially digested three-way lariat junctions (Supplementary Fig. 3A). Therefore, we removed reads mapping to a five nucleotide region around the brA ($-2$ to $+2$ around brA) to avoid confounding the analysis. Moreover, we established a way to visualize data across multiple experiments on a single plot for comparative purposes. The psiCLIP data provide nucleotide-resolution data on crosslinking that can be visualized with histograms, with the size of each bar corresponding to the number of crosslink events at each nucleotide (Fig. 3A). While nucleotide-resolution histograms convey precise positional information, as seen in the case of SmB data (Fig. 2), they cannot be used to present data across multiple experimental conditions on a single plot since overlapping bars would not be visible. Therefore, we present the data from helicase psiCLIP experiments as Gaussian smoothed curves with a window size of ten nucleotides, which reflects well the crosslinking trajectory (Fig. 3B), and is in agreement with the nine nucleotides of RNA bound to the two RecA domains in structures of DEAH-box helicases[7].

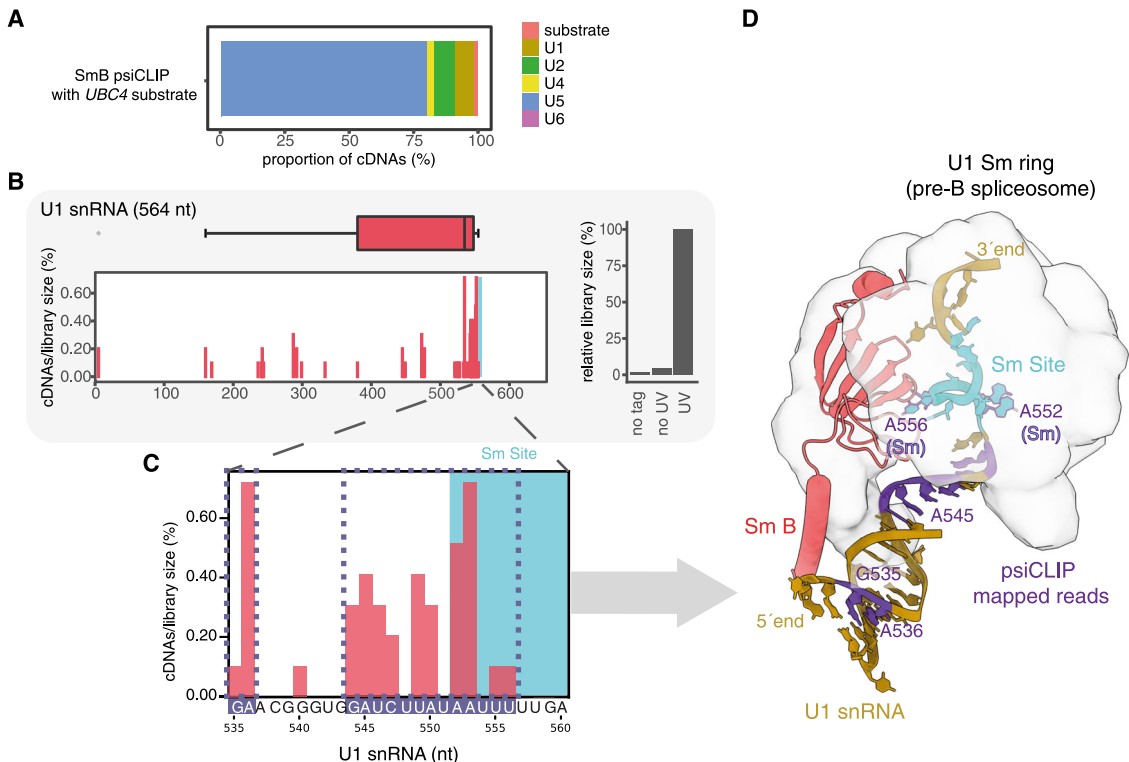

**Fig. 2 Validation of the positional specificity of psiCLIP with SmB. A** Proportion of cDNAs mapping to substrate and snRNAs for one SmB psiCLIP experiment on the short *UBC4* substrate. **B** Crosslinking to U1 snRNA. Crosslink events (−1 position of cDNA start) are represented as a histogram and box plot aligned to the indicated snRNA. On the histogram, the *y*-axis indicates the proportion of cDNAs out of all cDNAs mapped to snRNAs and pre-mRNA substrate. The box plot represents crosslink positions from one sample along the transcript, weighted by normalized cDNA count. The line across the box represents the median; the lower and upper bounds correspond to the first and third quartiles. The whiskers end at the largest and smallest value no further than 1.5 times the inter-quartile range. Outliers outside of this range are plotted as dots. The Sm site is highlighted in blue throughout. The bar chart represents the proportion of cDNAs in samples, normalized to the UV condition, which is shown as 100%. No tag: UV-irradiated untagged SmB, no UV: SmB-3× FLAG without UV irradiation, and UV: UV-irradiated SmB-3× FLAG. **C** Zoom in to show crosslinking specifically at the Sm sequence element on U1 snRNA, which is highlighted with a blue background. The U1 snRNA region highlighted in purple is shown in **D** mapped onto the cryo-EM structure. **D** Structure of the U1 snRNP Sm ring in the spliceosomal complex pre-B (ref. [40]; PDB: 5ZWN) with the SmB protein in salmon and the nucleotides of U1 snRNA in gold, with those showing the highest psiCLIP signal in purple or with a purple outline, corresponding to the purple boxes in **C**.

We performed Prp16 psiCLIP using the purified C-complex spliceosome, which was stalled and enriched using three different approaches: mutation of the 3′-SS from the canonical 3′-UAG| to 3′-UAC| (where | marks the splice site), addition of recombinant, ATPase-deficient Prp16-G378A mutant[44], or a combination of 3′-SS and Prp16 mutations. To assess substrate-specific effects, we compared *UBC4* and *ACT1*—two pre-mRNAs that have been used extensively in previous spliceosome functional and structural studies. Under similar conditions that were used to pull down Prp16 after UV crosslinking, Prp16 binds the lariat intermediate but not the pre-mRNA, demonstrating that our method captures reads derived from spliceosomes after branching (Supplementary Fig. 4A).

psiCLIP of Prp16 shows crosslinks predominantly on the pre-mRNA substrate, with little signal on snRNAs (Fig. 3C). The snRNAs that are not present in C complex, such as U1 and U4 snRNA, are under-represented compared with U2 and U5 snRNA, which are part of C complex (Fig. 3D). Our three stalling methods were expected to enrich for the same spliceosomal state—the C complex stalled right after branching. Indeed, we found that Prp16 makes similar RNA contacts regardless of the stalling method, demonstrating the high reproducibility of the psiCLIP method (Supplementary Figs. 3E, F and 4C, E). Nonetheless, we found that poorer crosslinking

efficiency for Prp16, compared to Prp22, for example, resulted in increased stochastic variation in the crosslinking profiles between replicates (Supplementary Fig. 4B–G).

On both splicing substrates, the binding pattern of Prp16 is widely spread, producing prominent crosslinks up to 40 nucleotides downstream of the brA and extending well beyond the predicted 9 nucleotides for the occluded site of a DEAD/H-box helicase[7]. On *ACT1* Prp16 shows a prominent peak ~30 nucleotides downstream of the brA, whereas on *UBC4* the main peak occurs 20 nucleotides downstream of the brA, though binding spans the entire region between the brA and the second exon for both substrates (Fig. 3E, F—sum of all replicates, and Supplementary Fig. 4C, E—three replicates shown separately). Although the position of the main crosslinking peak is significantly different between the two splicing substrates ($p < 0.001$, alpha = 0.05, Student's unpaired two sided *t* test, Supplementary Fig. 4H, I), comparisons between substrates need to be interpreted with caution, as crosslinking efficiency may be affected by the underlying RNA sequence. Our findings include positions previously identified using photo-reactive nucleotides[9,25] and confirm inferences from the C-complex cryo-EM structure, in which at least 18 nucleotides of RNA were predicted to span the distance between the brA and the RNA entry site of Prp16. This prediction was drawn from the six

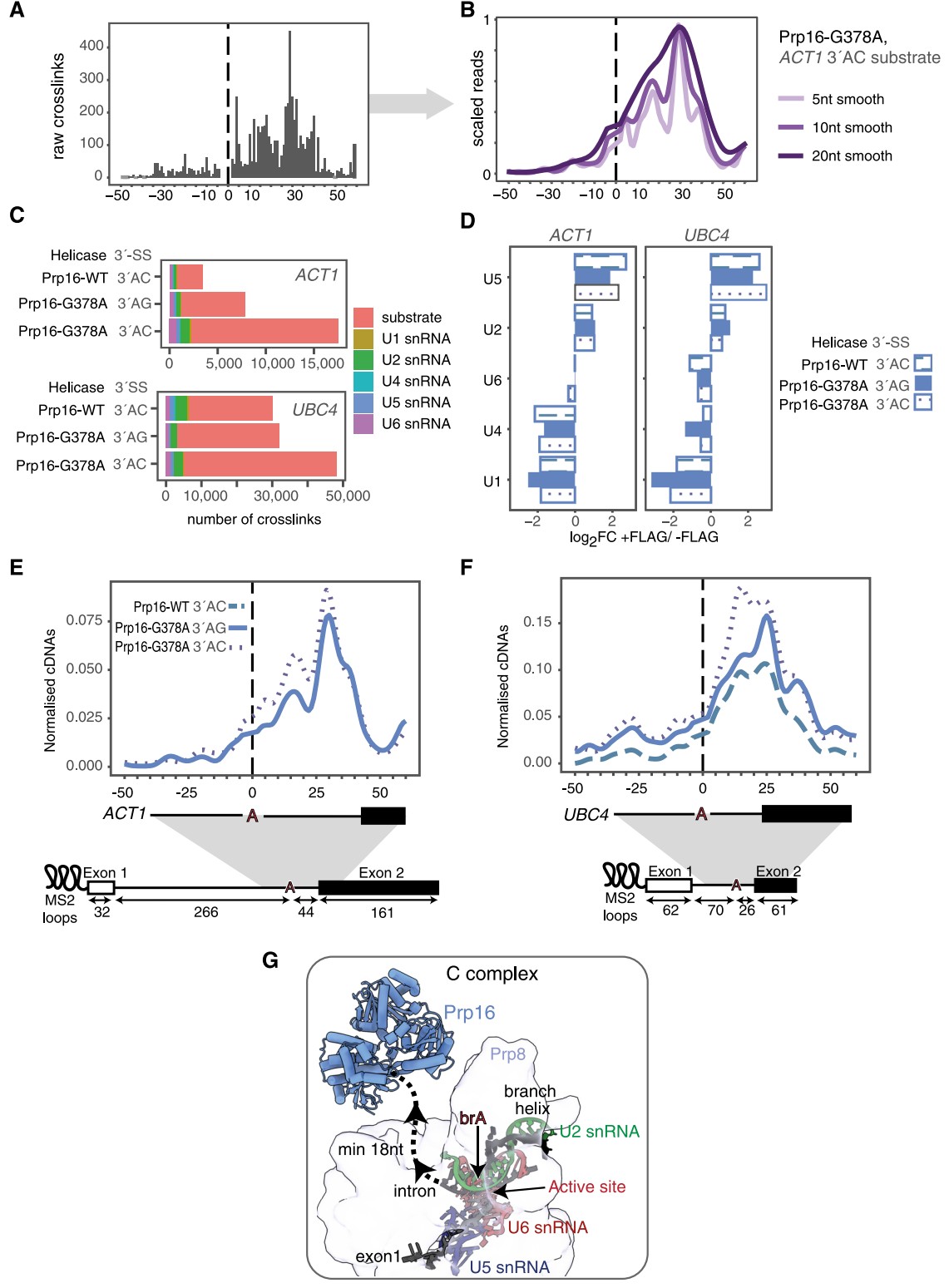

nucleotides closest to the brA, which could be modelled into ordered density, with the additional 12 added based on the approximate length that would be needed to reach the RNA entry site of Prp16 (ref. [8]; Fig. 3G). Importantly, Prp16 binding did not appear strictly constrained by the 3'-SS, as binding could be detected in the second exon for *UBC4*. Overall psiCLIP provides comprehensive positional information for Prp16 binding, showing an extensive binding profile in C complex, downstream of the brA.

**Prp16 binding depends on the brA to 3'-SS distance.** To test the constraints on Prp16 binding, we designed transcripts based on *ACT1* with increasing lengths between the brA and 3'-SS. Randomized 20 and 40 nucleotide sequences with the same GC content as the *ACT1* intron were inserted after the brA, creating transcripts that spliced with similar efficiency to the original transcript in a wild-type Prp16 background (Fig. 4A–C). psiCLIP was performed with C-complex spliceosomes assembled on these substrates and stalled using the ATPase-deficient Prp16 mutant

**Fig. 3 Prp16 psiCLIP data show substrate-specific binding in a region downstream of the brA. A**, **B** An example of Prp16 data shown as raw crosslinks and after Gaussian smoothing using different window sizes. Raw crosslink signal is shown as a grey histogram. **C** Total number of crosslinks summed across three replicates for each condition (UV crosslinked, 3× FLAG-tagged samples). Colour denotes the RNA species to which the crosslinks map. **D** Enrichment of snRNAs in FLAG-tagged Prp16 vs. untagged samples, represented as $\log_2$ fold change. Note that for this analysis, crosslinks were first normalized to the total number of snRNA crosslinks in each sample. **E**, **F** Mapping of Prp16 psiCLIP data onto *ACT1* and *UBC4* splicing substrates. Three replicates were summed. The smoothed lines show the truncation events normalized to endogenous yeast RNA, with the control signal subtracted from the UV signal. Lines were Gaussian smoothed with a window size of ten nucleotides. Positions along the transcript are shown relative to the brA. Crosslinking signals derived from the three different stalling methods show the same pattern. The main peak that reflects the position of the Prp16 helicase is slightly further downstream on *ACT1* than on *UBC4*. Note that wild-type Prp16 binding on 3′-AC *ACT1* substrate is excluded here due to small library sizes and low reproducibility (Supplementary Fig. 4F). **G** Cryo-EM structure of Prp16 in C complex shows the helicase at a distance from the brA (ref. [8]; PDB: 5LJ3). Six nucleotides were built into the density after the brA (brA +6 is the last ordered nucleotide). The predicted path of the intron and second exon is shown as a dotted line, implying a minimum distance of 18 nucleotides between the brA and the entry site of Prp16.

(Prp16-G378A). As the distance between brA and 3′-SS was increased, the binding region of Prp16 expanded to fill the full intronic region between the brA and 3′-SS (Fig. 4E–G). In all cases, a clear drop of signal was observed at the intron–exon boundary (Supplementary Fig. 5A–C), even though the normalized crosslinking signal generally decreases on the extended transcripts (Supplementary Fig. 5E). We observed crosslinking ~30 nucleotides upstream of the brA, but this was disregarded as it was inconsistent between replicates and it was also detected in the control samples (Fig. 4G, H). Thus, our results indicate that Prp16 binds downstream of the brA on the entire length of accessible single-stranded RNA, and does not strictly discriminate between introns and exons.

Prp16-mediated remodelling of C complex removes the brA from the active site to allow 3′-SS docking for exon ligation. Prp16 action thus results in remodelling of the branch helix, which forms between U2 snRNA and the substrate region upstream of brA (Fig. 3G). The branch helix may be disrupted during Prp16 action and such disruption may allow Prp16 to promote use of alternative 3′-SS[9]. A long-standing puzzle has been how Prp16, transiently located at the spliceosomal periphery, can remodel the branch helix, which is buried within the spliceosome core. Previous work suggests Prp16 can act indirectly without translocating through the branch helix[8,9]. To investigate whether Prp16 could disrupt RNA helices during remodelling of the spliceosome, we introduced a 40 nucleotide stem–loop into the *ACT1* intron between the brA and the 3′-SS (Fig. 4D, H, I and Supplementary Fig. 5D). This substrate was spliced in a wild-type Prp16 background, indicating that the Prp16 action is not appreciably hindered by the inserted sequence (Fig. 4D). We then performed psiCLIP on C complexes assembled on this substrate and stalled with the Prp16-G378A mutant. We observed Prp16 crosslinking mainly on the predicted single-stranded region between the stem–loop and the 3′-SS (Fig. 4H, I). The 5′ arm of the stem–loop showed very little crosslinking and the 3′ arm showed reduced crosslinking, suggesting that the stem–loop sequence indeed formed a secondary structure that hindered Prp16 binding. Since RBPs preferentially crosslink to single-strand RNA[45], the small amount of crosslinking observed towards the 3′ end of the 3′ arm (Fig. 4I) suggests that the Prp16 helicase mutant, which retains residual ATPase activity, may have partially unwound the stem–loop. Nonetheless, the impaired Prp16 binding we observe is consistent with a model in which Prp16 remodels the branch helix without translocating fully through the brA[9]. Although future experiments will be necessary to determine whether wild-type Prp16 could fully disrupt such stem–loop sequences, our findings for the mutant Prp16 helicase demonstrate that psiCLIP can be used to test hypotheses about the activity of DEAH-box helicases in the context of intact native spliceosomes, thus providing complementary functional assays to smFRET and cryo-EM studies.

**psiCLIP of Prp22 detects repositioning of the helicase after exon ligation.** During remodelling of C complex to the C* complex, Prp16 dissociates from its binding site on Prp8, allowing Prp22 to replace Prp16 on Prp8 and to stabilize the C* complex for exon ligation[10–12] (Figs. 1A, 3G, and 5A). Exon ligation forms the spliceosomal P complex that still contains both the excised intron and the mature mRNA. RNA density observed in the Prp22 helicase core in the cryo-EM structure of the C* complex was assigned to the second mRNA exon, with ~16 nucleotides needed to span the distance between the last nucleotide of the first exon and the centre of the Prp22 RNA-binding pocket[10]. However, biochemical experiments suggested Prp22 binds on the intron close to the 3′-SS before exon ligation[24]. In the structure of the P-complex spliceosome Prp22 was positioned between nucleotides 14–21 on the second exon of the *UBC4* transcript[12], consistent with biochemical experiments[18]. Thus the precise position of Prp22 on the RNA substrate before and after exon ligation remained unclear.

We performed Prp22 psiCLIP using multiple splicing substrates after enriching for spliceosomes stalled either before (C* complex) or after exon ligation (P complex). We enriched for the C* complex by assembling spliceosomes on *UBC4* and *ACT1* transcripts, in which the 3′-SS G(−1) ribonucleotide was replaced with a 2′-deoxynucleotide (dG), which blocks exon ligation[46]. psiCLIP was then performed using both the wild-type and the dominant-negative Prp22-K512A ATPase-deficient mutant. Again, there was little signal on snRNAs, with underrepresentation of U1 and U4 snRNAs, which are absent in C* complex (Fig. 5C, D). Prp22 binding covered a broad region downstream of the brA on both *UBC4* and *ACT1* transcripts. Crosslinks were detected throughout the intron between the brA and the 3′-SS, including a site at eight nucleotides before the 3′-SS, which was previously observed in splicing extracts for a substrate that could not undergo exon ligation[24]. Crosslinks were also detected in the second exon, with a major peak at 10–25 nucleotides downstream of the 3′-SS. The psiCLIP profile for Prp22 covers similar positions as those shown previously to crosslink to Prp8 and Prp22 using site-specific crosslinking in spliceosomes at the C*-complex stage[47,48], further validating our method. Thus Prp22 binds broadly on both the intron and second exon of the lariat intermediate before exon ligation. Importantly, both the wild-type and mutant Prp22 show a similar psiCLIP profile, confirming that RNA binding is ATP-independent, as suggested previously from cryo-EM studies, where RNA density was observed bound to an open, ATP-free, conformation of the helicase[10,49] (Fig. 5E, F and Supplementary Fig. 6A).

To determine if exon ligation changes Prp22 binding, we performed Prp22 psiCLIP in P-complex spliceosomes, which was enriched by adding recombinant Prp22-K512A dominant-negative mutant protein to block release of the mRNA. P-complex psiCLIP reads were enriched for mRNA junctions, with 10–15% of mapped

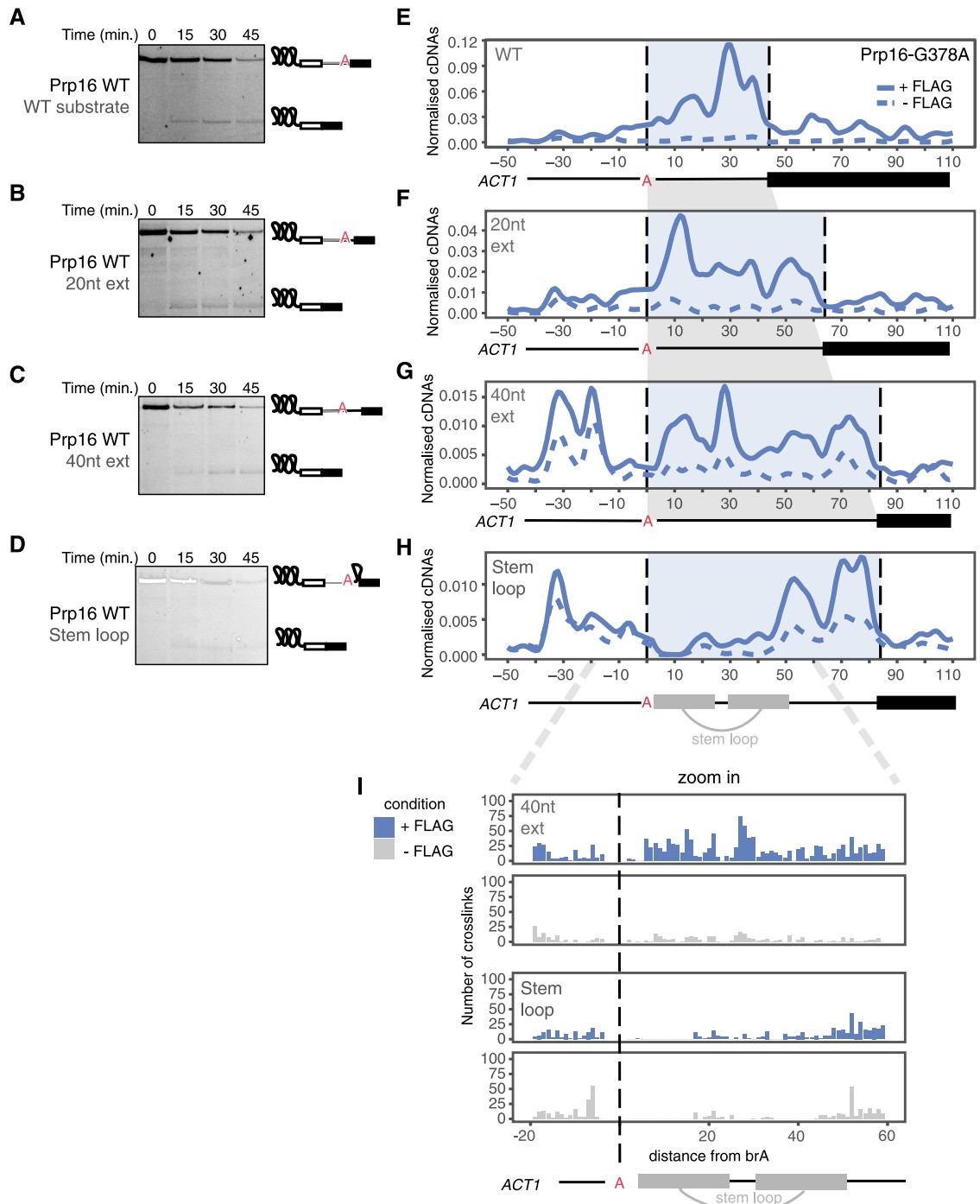

**Fig. 4 Prp16 binds the entire available single-stranded RNA between brA and the 3′-SS in *ACT1* extended substrates. A–D** All versions of the extended *ACT1* substrate are capable of being spliced. Splicing assays were performed in a Prp16-WT background. Only splicing products containing the second exon are visible, as the 3′ end of the transcript is fluorescently labelled. Gels are representative of two biological replicates. **E–H** Mapping of the psiCLIP data generated using Prp16-G378A mutant protein and various forms of the *ACT1* splicing substrate, including artificial extensions between the brA and the 3′-SS. The splicing substrates are aligned onto the brA, and the distance between the brA and the 3′-SS is highlighted in light blue. The smoothed lines show the truncation events normalized to endogenous yeast RNA, with the untagged signal subtracted from the tagged signal. Lines were Gaussian smoothed with a window size of ten nucleotides. Positions along the transcript are shown relative to the brA. The crosslinking signal spreads across the whole length of the sequence between the brA and 3′-SS. **I** Zoom in on the region around the brA shows the almost complete loss of crosslinking downstream of brA in the stem–loop experiment.

reads containing the canonical splice junction, compared with a maximum of 0.09% junction reads for the C* complex (Supplementary Fig. 6B). We found sharp binding peaks in the second exons of both splicing substrates at +10 and +20 nucleotides downstream of the exon–exon junction in *UBC4* and

*ACT1*, respectively, with binding extending as far as +40 nucleotides (Fig. 5G, H and Supplementary Fig. 6A). The binding region determined by psiCLIP is in perfect agreement with the binding site implied for *UBC4* by the complex P cryo-EM structure, and contains positions +10, +17, +22, and +35 on the

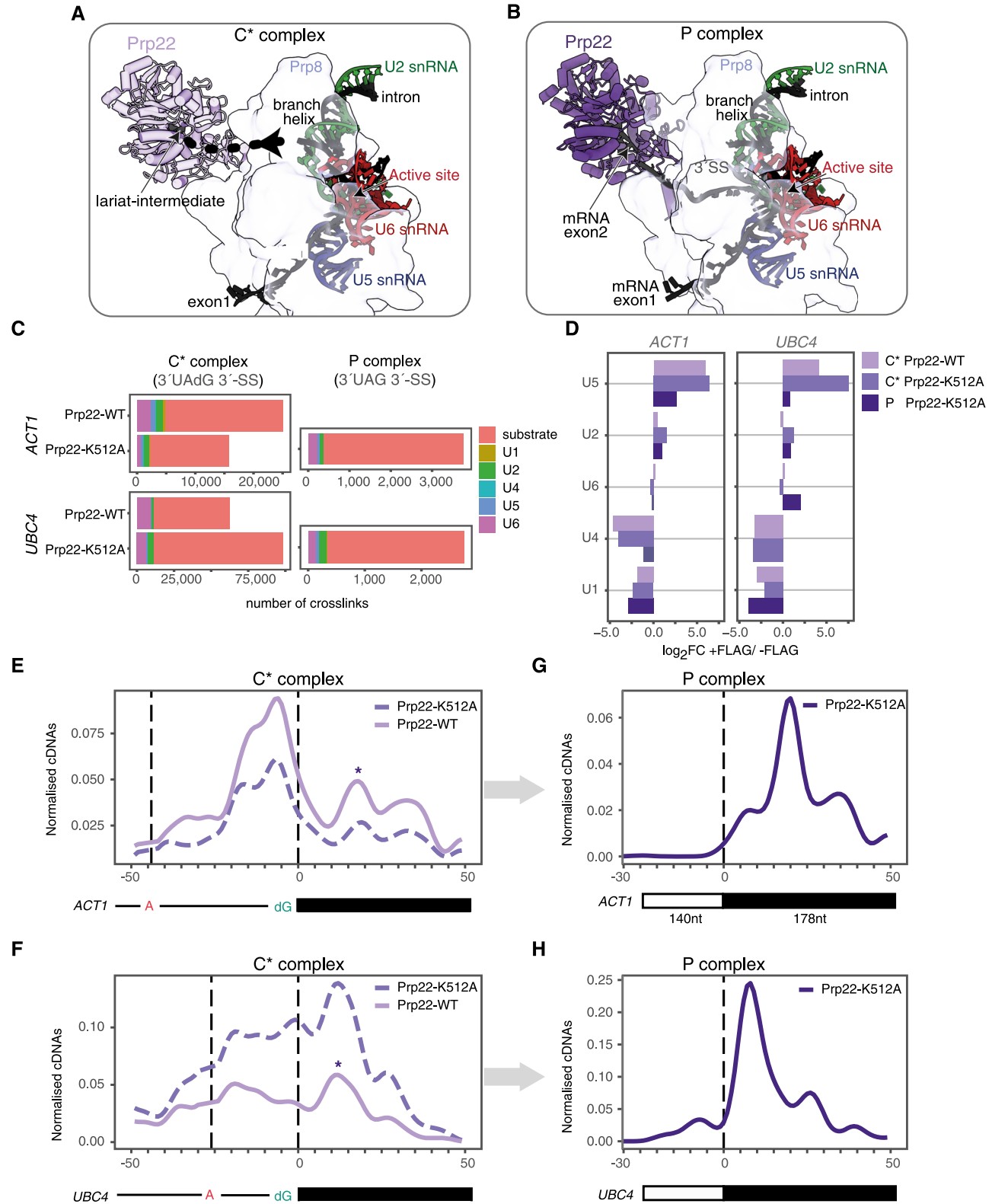

second exon of *ACT1*, which were previously identified by site-specific crosslinking[18]. Importantly, the major psiCLIP peak observed for Prp22 in complex P on the second exon is also observed in complex C* (Fig. 5E–H), indicating that Prp22 binds at similar positions within the second exon both before and after exon ligation. In contrast, psiCLIP suggests that Prp22 no longer binds the intron after exon ligation, in the P complex.

**psiCLIP of Prp22 is sensitive to the presence of auxiliary factors.** Exon ligation of *ACT1* and *UBC4* requires the exon-ligation factors Slu7 and Prp18 (ref. [50]). It has been suggested that Slu7 binds the spliceosome first, followed by Prp18 and then Prp22, although Slu7 and Prp18 may bind together as a heterodimer[43,50]. In the C* and P complexes, Slu7 and Prp18 bind the Prp8 RH domain to stabilize the exon-ligation

**Fig. 5 Prp22 psiCLIP indicates a shift from broad binding in C\* complex to narrow binding in P complex. A, B** Cryo-EM structures of C\* complex and P complex show Prp22 helicase in light and dark purple, respectively (refs. [8,12]; PDB: 5MQ0; PDB: 6EXN). Differential colouring for Prp22 is used simply for representation purposes; the helicase binds the same site in both complexes. The dotted line represents a possible path for unbuilt RNA regions. **C** Total number of crosslinks summed across two UV crosslinked, FLAG-tagged replicates for each condition. Colour denotes the RNA species to which the crosslinks map. **D** Enrichment of snRNAs in FLAG-tagged Prp16 vs. untagged samples, represented as log₂ fold change. Note that for this analysis, crosslinks were first normalized to the total number of snRNA crosslinks in each sample. **E–H** Mapping of Prp22 psiCLIP data onto *ACT1* and *UBC4* transcripts. The smoothed lines show the truncation events normalized to endogenous yeast RNA, with the control signal subtracted from the UV signal. Crosslinks are shown after Gaussian smoothing with a ten nucleotide window. Positions on the substrate are given relative to the 3′-SS. P-complex crosslinks are shown with the intron mapping reads removed. **E** Prp22-WT and Prp22-K512A psiCLIP on *ACT1* substrate in C\* complex. **F** Prp22-WT and Prp22-K512A psiCLIP on *UBC4* substrate in C\* complex. In **E** and **F**, the proposed P-complex-like signal, potentially diagnostic of 3′-SS docking, is indicated with an asterisk (\*). **G** Prp22-K512A psiCLIP on *ACT1* substrate in P complex. **H** Prp22-K512A psiCLIP on *UBC4* substrate in P complex.

conformation[10,12] (Fig. 6A) and a loop from Prp18 binds near the 3′-SS to stabilize its docking in the active site[12]. Prp18 was also suggested to promote Prp22 binding to the spliceosome[50], yet the structures of the C\* and P complexes did not identify direct contacts between Prp18 and Prp22. We therefore sought to investigate whether Prp18 affects Prp22 binding patterns by performing psiCLIP upon depletion of Prp18 from splicing extracts. We also compared wild-type Prp22 and ATPase-deficient Prp22-K512A to assess how the Prp22 binding pattern depends on its ATPase activity.

As expected, the Prp18-depleted extract accumulates lariat intermediate with both the wild-type and mutant Prp22, indicating a defect in exon ligation, which could be rescued by adding recombinant Prp18 (Supplementary Fig. 7A). Next, psiCLIP was performed using *ACT1* transcripts containing either the canonical 3′-SS or the suboptimal dG modified 3′-SS (3′-UAdG), which in the absence of Prp18 would both be expected to enrich for spliceosomes stalled at the C\* stage. Indeed, purified complexes contained lariat intermediate and thus appeared stalled after branching (Supplementary Fig. 7B, C). We observed a striking difference in the substrate binding of wild-type and mutant Prp22 helicases in Prp18-deficient spliceosomes. Wild-type Prp22 displayed very little psiCLIP signal in the Prp18-deficient spliceosomes (Fig. 6B–D and Supplementary Fig. 7D–F), whereas the ATPase-deficient Prp22 helicase showed similar binding in the presence or absence of Prp18 (Fig. 6B–D). In contrast to wild-type Prp22, the ATPase-deficient mutant Prp22-K512A allows more exon ligation at the canonical splice site, as indicated by the presence of more spliced mRNA reads in the psiCLIP samples (Supplementary Fig. 7G). Nonetheless, the majority of the psiCLIP signal derives from lariat intermediates, suggesting that the depletion of Prp18 destabilizes binding of wild-type Prp22 before exon ligation, in agreement with previous biochemical data[50]. Overall, our psiCLIP data indicate that ATP hydrolysis destabilizes Prp22 binding when the optimal composition of the spliceosome is disrupted.

## Discussion

Splicing fidelity is safeguarded by eight ATP-dependent helicases, but their precise binding sites and dynamic interactions with snRNAs and pre-mRNAs remain poorly understood. Detailed knowledge of helicase binding profiles is required to understand how they promote conformational changes in the spliceosome and ensure splicing fidelity. We developed psiCLIP to define the RNA contacts of two of the most elusive DEAH-box helicases: Prp16 and Prp22. We establish new data normalization and visualization approaches to present insights from comparative experiments upon mutation of these helicases or depletion of auxiliary factors in various defined spliceosomal states. Our study thus presents a strategy that can complement functional insights from structural studies by combining biochemical isolation of

defined conformations of large RNA–protein complexes with high-resolution interactomics (psiCLIP) and bioinformatics.

psiCLIP provided mechanistic insights into helicase activity by revealing the complete binding profile of the helicases Prp16 and Prp22 on their native substrates, in defined spliceosome complexes. DEAH-box helicases like Prp16 and Prp22 were generally proposed to bind away from their targets and hydrolyse ATP to translocate through RNA structures in the 3′ to 5′ direction[51]. However, cryo-EM structures of the C, C\*, and P complexes showed that Prp16 and Prp22 bind at the periphery of the spliceosome at least 15–20 nucleotides away from the RNA structures that they remodel: the branch helix and the mRNA (or lariat intermediate), respectively. Thus both Prp16 and Prp22 appeared unable to remodel these sites by directly translocating through them. In agreement, we found extensive binding of both Prp16 and Prp22 on the substrate downstream of the sites that they remodel, supporting a mechanism of action at a distance rather than processive translocation through the remodelled RNA sites.

Single-molecule FRET studies suggested that Prp16 translocates on the lariat intermediate, while remaining anchored on the C complex (Fig. 3G), acting as a molecular winch that pulls the substrate out of the active site and disrupts the branch helix to allow remodelling[9]. Our psiCLIP data provides further support for this winching model. Instead of binding at a fixed distance from the brA, we found that Prp16 binds over the full available single-stranded RNA region between the brA and 3′-SS, even when this distance is extended with unstructured RNA. Nonetheless, most of the Prp16 psiCLIP signal occurred within ~45 nucleotides downstream of the brA, which correlates with the most common distance between brA and 3′-SS for unstructured yeast introns[52]. This defined region of accessible RNA is likely determined by the binding of Prp16 to spliceosomal components, such as Prp8 (Fig. 3G), as well as by RNA secondary structures formed between the brA and the 3′-SS. Additional single-molecule and chase experiments are necessary to determine if the broad binding we observed reflects movement during translocation or other forms of dynamic contacts, such as binding, release, and re-binding. However, since broad contacts were detected for both the wild-type helicase and the ATPase mutant, which is likely impaired in translocation[44] (Fig. 3E, F and 4E–G), we propose that the broad binding we observe reflects, at least in part, multiple rounds of binding and dissociation by one or more helicase molecules.

Our Prp16 psiCLIP data shows that Prp16 binding can extend into unstructured RNA inserted after the branch helix. By contrast, an ATPase-deficient Prp16 mutant could not appreciably bind into a structured stem–loop region, suggesting that it cannot disrupt strong RNA helices, even though the structured stem–loop allowed splicing, and thus C to C\* remodelling. Our data thus provide support for a mechanism of action at a distance, in which Prp16 winches towards the branch helix but remodels the spliceosomal active site without translocating through the branch helix.

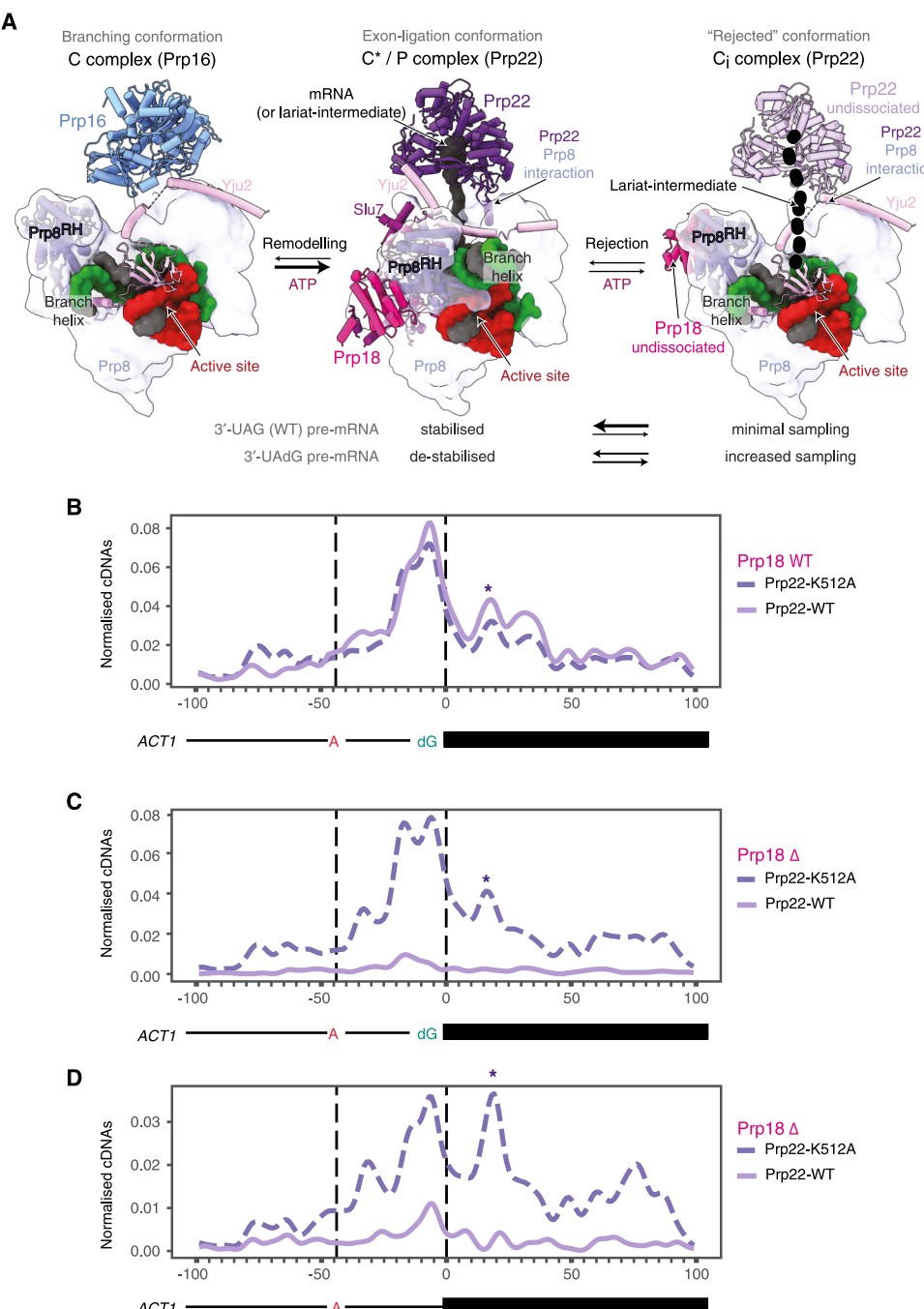

**Fig. 6 Prp18 depletion destabilizes wild-type Prp22 binding on the pre-mRNA substrate, but not the binding of the ATPase-deficient Prp22 mutant.**
**A** Binding of Prp22 in the exon-ligation conformation in C* (ref. [10]; PDB: 5MQ0) and P (ref. [12]; PDB: 6EXN) complexes is stabilized by Prp18 and Slu7 (middle panel). For comparison, binding of Prp16 is shown in the C complex (left panel). During rejection by Prp22, the spliceosome may revert to a C-like conformation ($C_i$, ref. [43]; PDB: 7B9V), in which both Prp18 and Prp22 may remain loosely bound (right panel). **B** Prp22 binding to spliceosomes assembled on the suboptimal 3'-UAdG substrate in mock-depleted extracts. **C** psiCLIP of Prp22 in spliceosomes assembled on the 3'-UAdG substrate following depletion of Prp18 (Prp18Δ). **D** psiCLIP of Prp22 in spliceosomes assembled on the wild-type ACT1 substrate following depletion of Prp18. Both wild-type and ATPase mutant Prp22 are shown for all experiments. The proposed P-complex-like signal, potentially diagnostic of 3'-SS docking, is indicated with an asterisk (*). In all panels, crosslinks are shown normalized to endogenous yeast RNA, with reads from untagged Prp22 subtracted from tagged Prp22 reads, and after applying Gaussian smoothing with a ten nucleotide window.

Similarly to Prp16, psiCLIP revealed broad binding of Prp22 downstream of the brA before exon ligation, independently of ATPase activity. The high reproducibility of binding profiles observed at specific substrate positions in several replicates and across stalling conditions (Supplementary Fig. 4B–G) likely represents an average of interactions in defined complexes, rather than resulting from heterogeneous binding or processive translocation events across multiple complexes. Thus, we interpret the broad binding as evidence that the helicases can bind at multiple defined regions along the substrate. In the C* complex, we detected Prp22 binding on the intron before the 3'-SS and on the second exon downstream of the 3'-SS. The strong signal observed for the

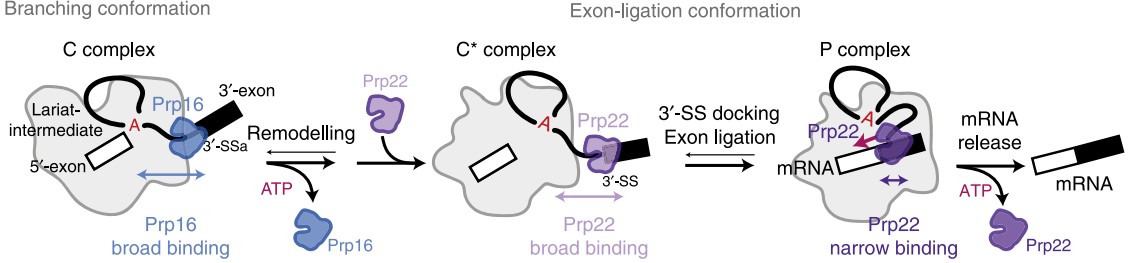

**Fig. 7 Model for substrate binding by spliceosomal helicases during the catalytic stage of pre-mRNA splicing.** After branching, Prp16 binds the lariat intermediate broadly downstream of the brA, though the main binding peaks are observed between the brA and the 3′-SS. Prp16 dissociates during remodelling to the C* conformation, in which it is replaced by Prp22 before docking of the 3′-SS. In complex C*, before exon ligation, Prp22 binds the lariat intermediate broadly on the 3′-exon and the intron. Such broad binding may facilitate 3′-SS docking to allow exon ligation. After exon ligation, in complex P, Prp22 engages the mRNA in a narrow binding window on the 3′-exon, from which it pulls on the RNA substrate to release the mRNA and dissociates following mRNA release.

wild-type Prp22 protein, as well as the broad profile seen for the ATPase-deficient K512A mutant, which is unable to translocate, suggest multiple rounds of stochastic binding and dissociation, rather than a single ATP-dependent translocation and dissociation event. Notably, the psiCLIP-binding profile is maintained upon the inhibition of Prp22 ATPase activity, consistent with an ATP-independent role for Prp22 in exon ligation[15].

The broad accessibility for Prp22 binding in the intron on the 3′-UAdG substrate indicates that the 3′-SS is not stably docked in the active site in a significant population of C* spliceosomes assembled on this substrate. Indeed, the 3′-UAdG substrate does not produce mRNA and no stably docked 3′-SS was detected in the cryo-EM structure of the C* complex[10]. The extended intron binding profile in the C* complex suggests that in the absence of a stably docked 3′-SS Prp22 can translocate along the lariat-intermediate upstream of the 3′-SS, or bind and rebind in this region, potentially to reject these spliceosomes, thus ensuring that exon ligation occurs only at the correctly docked 3′-SS. Supporting this idea, recent cryo-EM studies indicate that a population of spliceosomes assembled on the 3′-UAdG substrate revert to the branching conformation, potentially as a result of rejection of the C* complex by Prp22 (refs. [10,43]).

psiCLIP also detected substrate rearrangements resulting from substrate docking and catalysis. In complex P, Prp22 binding shifted to a narrower region 10–25 nucleotides downstream of the 3′-SS, and we detected the same major peak in the 3′-exon in both complex C* and complex P (Fig. 5G, H). This exonic binding position is likely associated with a stably docked 3′-SS and consequently this 3′-exon peak may be diagnostic for complexes in the exon-ligation conformation. Given the 3′ to 5′ direction of helicase movement on RNA, the observed shift in the Prp22-binding profile from major peaks in the intron in C* to major peaks in the 3′-exon in P complex is inconsistent with a simple mechanism of processive 3′ to 5′ translocation during Prp22 action. Instead, we propose that Prp22 may initially engage the substrate on the 3′-exon before exon ligation. After exon ligation, Prp22 would thread the substrate from the 3′-exon to release the mRNA from the spliceosome, potentially through repeated binding and dissociation (Fig. 7 and Supplementary Fig. 8A). Before exon ligation, Prp22 may promote, and proofread, 3′-SS sampling and docking by dynamic engagement with the lariat intermediate, as evidenced by binding in the intron upstream of the 3′-SS (Figs. 5E, F and 7). Consistent with this idea, Prp22 was shown to promote sampling of alternative 3′SS (ref. [9]).

The psiCLIP-binding profile may thus reflect the proofreading activity of Prp22 and could be influenced by factors that stabilize the P complex. Indeed, we found that Prp22 binding to the

substrate depended on the exon-ligation factor Prp18, which promotes 3′-SS docking. Prp18 indirectly stabilizes Prp22 binding by locking Prp8 in the exon-ligation conformation in the C* and P complexes (Fig. 6A). As expected, the Prp22 ATPase-deficient K512A mutant remained bound to both the canonical and sub-optimal 3′-UAdG substrate upon Prp18 depletion, consistent with the ATPase-deficient mutant being intrinsically less prone to dissociation, even when the exon-ligation conformation is destabilized in the absence of Prp18 (Supplementary Fig. 6G and Fig. 6A). Strikingly, binding by the wild-type Prp22 was significantly reduced in the absence of Prp18. In the absence of Prp18, Prp22 may dissociate from the spliceosome and may not rebind. Depletion of Prp18 may also indirectly increase the ATPase activity of Prp22, promoting dissociation and reducing stable binding to the substrate. Indeed, structural studies have shown that the absence of Prp18 destabilizes the C* and P conformation[43], and would likely weaken Prp22 binding onto Prp8, which may be necessary for stable association with the substrate. Similar to Prp22's proofreading activity for correct 3′-SS selection[14], wild-type Prp22 may not bind stably in the absence of Prp18 and could therefore prevent potentially erroneous exon ligation by incompletely assembled spliceosomes. Consistent with this idea, in the presence of Prp22-K512A we detected exon–exon spanning reads corresponding to mRNA (Suplementary Fig. 7G). Thus, when Prp22 ATPase activity is compromised splicing can proceed in the absence of Prp18, suggesting that the ATPase activity of Prp22 may reject spliceosomes lacking Prp18.

Our data imply that productive Prp22 association with the substrate is dependent more broadly on the stability of the exon-ligation conformation (Supplementary Fig. 8). In this proofreading model, Prp22 monitors the thermodynamic stability of the spliceosome by discriminating against complexes that cannot complete exon ligation before Prp22 action, whether due to a suboptimal, non-canonical, 3′-SS or a suboptimal spliceosome, resulting for example from poor binding of auxiliary factors. For such suboptimal spliceosomes the exon-ligation conformation would be destabilized. Before exon ligation can occur, Prp22 bound initially in the 3′-exon may either translocate from this position or dissociate and rebind upstream of the 3′-SS to reject lariat intermediates that cannot stably reach the exon-ligation conformation (Fig. 5E, F and Supplementary Fig. 8B). Dissociation of Prp22 would reject these complexes and may cause the spliceosome to revert from the C* complex to an intermediate C-like conformation ($C_i$). This $C_i$ conformation may not allow stable re-binding by Prp22 (Fig. 6A), thus preventing progress to the P complex and blocking exon ligation by suboptimal spliceosomes. Indeed, the $C_i$ conformation, in which Prp18 remains loosely bound, was

observed by cryo-EM for spliceosomes assembled on the sub-optimal 3′-UAdG substrate[43]. Depletion of Prp18 may further destabilize the C* and P complexes, and push spliceosomes towards such a C-like conformation. Thus, psiCLIP complements the interpretation of structural studies and provides mechanistic insights into the proofreading activities of spliceosomal ATPases.

Taken together, our study raises new questions about the activity of Prp22 and Prp16 ATPases and, more generally, provides valuable information about active and passive helicase activities. Further biochemical work will be required to fully elucidate the mechanisms behind the phenomena uncovered by psiCLIP. While psiCLIP is well suited for comparisons between conditions, the absolute crosslinking levels in psiCLIP are affected by the variable crosslinking efficiencies of nucleotides and amino acids, which can lead to biases at the sequence level, as in all techniques utilizing UV crosslinking. It is known that uridine is the most favourable crosslinker, followed by guanosine, cytosine, and adenosine, respectively[53]. Thus, comparisons of psiCLIP data between transcripts and/or proteins should take into account such sequence-specific biases. Looking to the future, we envision that psiCLIP can be extended to study helicases and other RBPs in additional biochemically defined RNP complexes, such as the RBP–rRNA interactions during ribosome biogenesis. For example, time-resolved psiCLIP could be used to track the binding profile of a helicase in motion during RNP remodelling. Thus, psiCLIP provides a versatile method to investigate fundamental mechanisms of RNP dynamics and remodelling.

## Methods

**Yeast strains**. All strains used in this study were derived from BCY123 (*MATa, can1, ade2, trp1, Ura3-52, his3, leu2-3, 112, pep4::his+, prb1::leu2+,bar1::HisG+, lys2::pGAL1/10-GAL4+*; Supplementary Table 1). In brief, sequences coding for protein tags and the respective resistance cassette were amplified from plasmids coding for the protein tag and resistance cassette with ~60–90 nucleotides homology to the end of the target gene and the beginning of its 3′ UTR (Supplementary Tables 1 and 3). The linear PCR product was transformed into BCY123 and the cells were plated on selective media containing either 100 µg/mL ClonNAT, 300 µg/mL hygromycin, or 250 µg/mL G418. Positive clones were verified by Sanger sequencing and expression of the tagged protein was analysed by western blot using monoclonal ANTI-FLAG® M2-Peroxidase antibody (Sigma Aldrich) with 1:3000 dilution, c-Myc antibody (9E10) HRP (Santa Cruz Biotechnology) with 1:1000 dilution, or HA-probe antibody (F-7) HRP (Santa Cruz Biotechnology) with 1:1000 dilution to detect the respective tag. Strains containing several C-terminal tags underwent the cycle several times.

For recombinant protein expression, plasmids based on pRS424 and pRS426 vectors carrying *TRP*1 and *URA3* markers, respectively, were transformed into BCY123 and grown on YM4 selective media (Supplementary Table 1). Both Prp16 and Prp22 constructs were encoded on pRS424 under the GAL-GAPDH hybrid promoter.

**Preparation of splicing substrates**. In vitro transcriptions were performed from pUC-based vectors with the desired sequence following the T7 promoter (Supplementary Table 2) and containing a 3× MS2 stem–loop sequence either at the 3′ or 5′ end[54]. Artificial extensions of the *ACT1* substrate were generated using oligonucleotides with a random sequence that maintained the same GC content as that of the original sequence between brA and 3′-SS. These were cloned into a restriction site following the brA position (Supplementary Table 3). Single clones were selected with either a single 20 nucleotide insertion (5′-CGA TTT TAT TTA TTT GAT CT-3′), a 40 nucleotide tandem insertion (5′-CGA AAA TCA AGA TAA ATA ATC GAA AAT CAA GAT AAA TAA T-3′), or a 40 nucleotide head-to-head insertion to generate an RNA stem–loop (5′-CGA TTA TTT ATC TTG ATT TTC GAA AAT CAA GAT AAA TAA T-3′). Long RNA pieces were generated by run-off transcription. Some splicing substrates (Supplementary Table 2), including substrates containing a 2′-deoxynucleotide, were generated by ligation from a long 5′ piece ending before the 3′-SS and short 3′ oligonucleotides. For the respective *UBC4* substrates, oligonucleotides carrying a 3′-Cy5 fluorophore (purchased from Sigma) were added by ligation (Supplementary Table 2). For *ACT1*, after ligation of an oligonucleotide and an RNA piece generated by run-off transcription, 3′ end labelling was performed by enzymatic ligation of pCp-Cy5 (Supplementary Table 2). To generate precise 3′ or 5′ ends for ligation, transcripts included a hepatitis delta virus or hammerhead ribozyme sequence at the respective end. To join RNA pieces by splint-mediated ligation, the 5′-ends were first phosphorylated with T4 PNK (New England Biolabs). RNAs were annealed to bridge DNA oligonucleotides complementary to the junction by slowly decreasing the temperature from 80 to 25 °C before ligation with T4 DNA ligase (Supplementary Table 3)[46]. Ligated transcripts were gel purified after each ligation step.

**Expression and purification of recombinant Prp16 and Prp22**. BCY123 cells were co-transfected with expression vectors pRS426 and pRS424, the latter coding for the recombinant protein (Supplementary Table 1). Positive transformants were grown in 24 L YM4 selective media supplemented with 1% raffinose at 30 °C. Protein expression was induced with 2% final concentration of galactose at OD600 1.0. After 12–16 h further growth at 30 °C, cells were harvested, resuspended in one volume 2× lysis buffer (2 M NaCl, 100 mM Tris-Cl pH 9.0, 2 mM imidazole, 20 mM β-mercaptoethanol (β-ME), 0.2% IGEPAL CA-630, 4 mM CaCl₂, 2 mM magnesium acetate, and EDTA-free protease inhibitor cocktail (Roche)), and frozen in liquid nitrogen in droplet form. Cells were disrupted in a 6870 Freezer/Mill (SPEX SamplePrep). After thawing, the pH of the extract was raised to 8.5 by addition of 1 M Tris base. Cell debris were removed by ultracentrifugation at 195,000 × g for 90 min. The supernatant was incubated with 2 mL Calmodulin-sepharose beads (home made) for 12–16 h at 4 °C. Beads were washed with 5 × 50 mL CAL wash buffer (500 mM NaCl, 20 mM Tris-Cl pH 8.0, 2 mM CaCl₂, 1 mM magnesium acetate, 1 mM imidazole, and 10 mM β-ME) and the proteins eluted in 10 × 5 mL CAL elution buffer (500 mM NaCl, 20 mM Tris-Cl pH 8.0, 2 mM EGTA, 1 mM magnesium acetate, 1 mM imidazole, and 10 mM β-ME). Protein-containing fractions were pooled and dialyzed against Ni-NTA binding buffer (1 M NaCl, 20 mM Tris-Cl pH 8.5, 5 mM imidazole, and 10 mM β-ME) for 4 h at 4 °C. After 14 h binding to 4 mL Ni-NTA agarose beads (Qiagen), the beads were first washed with 15 mL Ni-NTA binding buffer and then with 15 mL Ni-NTA wash buffer (1 M NaCl, 20 mM Tris-Cl pH 8.5, 15 mM imidazole, and 10 mM β-ME). The protein was eluted in about six 4 mL fractions with Ni-NTA elution buffer (1 M NaCl, 20 mM Tris-Cl pH 8.5, 250 mM imidazole, and 10 mM β-ME). Helicases were dialyzed against the respective helicase storage buffer for 4 h at 4 °C and stored at −80 °C until further use (20 mM HEPES pH 7.9, 0.2 mM EDTA, 0.5 mM DTT, 20% glycerol with 250 mM KCl for Prp16 and 300 mM KCl for Prp22).

**Expression and purification of recombinant Prp18**. The coding sequence for Prp18 was cloned into pET14. The protein was expressed as N-terminal His6-thrombin-Prp18 in *Escherichia coli* BL21 (DE3)RIL cells. The protein was first purified on Ni-NTA-agarose beads and peak fractions dialyzed against buffer containing 10 mM potassium phosphate and 50 mM KCl pH 7.4. The protein was further purified on a hydroxyapatite column, eluted with 0.5–3.0% ammonium sulfate, and dialyzed into a suitable buffer for splicing reactions (20 mM HEPES pH 7.9, 0.2 mM EDTA, 0.5 mM DTT, 20% glycerol, and 250 mM KCl).

**Splicing extract preparation and in vitro splicing**. Yeast splicing extract was prepared using the liquid nitrogen method[55]. In vitro splicing reactions were performed with 2.5 nM splicing substrate in 40% splicing extract, 2 mM ATP, 2.5 mM MgCl₂, and 60 mM potassium phosphate pH 6.5 (ref. [56]). The resulting RNA species were phenol extracted and analysed on 5% (for *ACT1*) or 10% (for *UBC4*) denaturing polyacrylamide gels. Quantification of gel bands was done using ImageJ (ref. [57]).

**Protein depletion from splicing extract**. Prp18-3× HA was depleted from splicing extract by increasing the KCl concentration to 750 mM and incubating twice with 1/10 volume anti-HA magnetic beads at 4 °C for several hours, followed by dialysis against 20 mM HEPES pH 7.9, 0.2 mM EDTA, 0.5 mM DTT, 20% glycerol, and 200 mM KCl for 16 h. To ensure complete depletion of Prp18 while preventing co-depletion of Slu7, western blots against Prp18-3× HA and Slu7-9xcmyc were performed. To rescue the splicing defect caused by Prp18 depletion, rPrp18 was added to the splicing reaction at a final concentration of 165 ng/mL (ref. [58]).

**Spliceosome assembly and purification**. Complexes were assembled in a 1.5 mL splicing reaction on the indicated pre-mRNA substrate, which was pre-bound to a 1.25 fold excess of MS2-MBP fusion protein[54]. For reactions containing recombinantly expressed helicase mutants, the splicing extract was firstly incubated with 15 ng/mL final concentration of dominant-negative mutant protein (Prp16-G378A or Prp22-K512A). Reactions were incubated at 23 °C for 30 min (*UBC4* transcripts) or 60 min (*ACT1* transcripts). For C and C* complex purification, the reaction mixture was adjusted to 2 mM glucose and incubation prolonged for 5 min. For the C* complex, beads were incubated with 2 mM ATP/2 mM MgCl₂ for 30 min at room temperature before the wash steps[10]. For P-complex purification, to remove spliceosomes before the P-complex stage, reactions were incubated with 5 µM of a DNA oligonucleotide complementary to the 3′-exon for an additional 20 min (ref. [12]; Supplementary Table 3).

For all spliceosome preparations, the reaction mixture was centrifuged through a 40% glycerol cushion in buffer A (20 mM HEPES, pH 7.9, 75 mM KCl, and 0.25 mM EDTA). The cushion was collected and applied to amylose resin. After 15 h of incubation at 4 °C, the resin was washed three times with 1 mL buffer A and eluted for 20 min in 200 µL buffer A containing 5% glycerol, 0.01% NP-40, and 12 mM maltose.

**Spliceosome crosslinking, immunoprecipitation, and cDNA library preparation**. For each helicase experiment, at least two replicates were produced from independently assembled and purified spliceosomes, although not necessarily from independent extract preparations. In addition, control samples containing no tag on the protein and non-irradiated samples were prepared. Crosslinking and immunoprecipitation were adapted from the original iCLIP protocol[34] with the following modifications. The 200 µL spliceosome eluate was irradiated by UV light using a Stratalinker 2400 at 254 nm. Spliceosomes with the target protein SmB were irradiated with $312 \times 100$ µJ/cm² and spliceosomes with the target protein Prp16 or Prp22 with $312 \times 100$ µJ/cm² (assembled on *UBC4*) or $625 \times 100$ µJ/cm² (assembled on *ACT1*). A total of 72 mg urea was added and dissolved by shaking to gain a 6 M solution to denature the complex. The protein of interest was captured by magnetic anti-FLAG beads. The beads were incubated in 20% S100, 8 mM EDTA, and 40 µL debranching buffer (20 mM HEPES 7.9, 20% glycerol, 100 mM KCl, and 0.5 mM DTT) for 15 min to debranch intron lariats[59]. A total of 0.5–5 units (depending on complex and splicing transcript) of RNase I was added over 3 min at 37 °C for fragmentation. Dephosphorylation, adapter ligation, radioactive labelling, isolation of the RNA–protein complex, and cDNA library preparation were performed as in iCLIP. cDNA libraries were sequenced at the Illumina HiSeq 2500 platform with cycle lengths between 100 and 250 nucleotides in single-read mode using HiSeq control software v2.2.68. An overview of all psiCLIP samples with respective adapter and RT primer sequences is given in Supplementary data 1.

**Data processing**. A custom psiCLIP software pipeline was developed in Snakemake[60] and is available from github.com/luslab/psiclip. The code is easily extended to other experimental scenarios and substrates. Reads were demultiplexed using iCount demultiplex and trimmed for quality using Trim Galore! (https://www.bioinformatics.babraham.ac.uk/projects/trim_galore/). Quality of sequencing data was assessed with FastQC (https://www.bioinformatics.babraham.ac.uk/projects/fastqc/). Trimmed reads were mapped to a custom transcriptome index consisting of *S. cerevisiae* U1, U2, U4, U5, and U6 snRNAs alongside the pre-mRNA substrate sequence, using STAR aligner[61]. Key parameters were: `–alignEndsType EndToEnd` to prevent soft-clipping of reads which would obscure the truncation site, `–outFilterMismatchNmax 2` to allow a maximum of two mismatches and `–seedSearchStartLmax 16`, which means the read is split into more seeds resulting in a potentially more sensitive search given our short reads. Subsequently, Sambamba was used to retrieve reads mapping only in the forward orientation to the custom transcriptome[62], as psiCLIP is stranded. Finally, the cDNA start position −1 is taken as the crosslink site and bed files are generated, using Bedtools and R[63]. Reads that did not map to the custom transcriptome index were then mapped to the yeast genome (SacCer3) and similarly processed to generate crosslinks. We used the proportion of yeast genome mapping RNA in the library as internal normalization, much like a spike-in[64]. Further plotting and analysis is performed in R using dplyr (https://cran.r-project.org/web/packages/dplyr/index.html), ggplot2 (ref. [65]), and smoother (https://CRAN.R-project.org/package=smoother) packages. Data for Prp22 and Prp16 are presented as Gaussian smoothed curves with a window size of ten nucleotides. In the main figures, the sum of all replicates is shown, where available, with replicates shown separately in the Supplementary Material.

## Data availability
psiCLIP data are available to download from ArrayExpress at E-MTAB-8895. Structural biology data that was used for comparison was taken from the following publicly available sources: https://doi.org/10.2210/pdb5ZWN/pdb, https://doi.org/10.2210/pdb5LJ3/pdb, https://doi.org/10.2210/pdb5MQ0/pdb, https://doi.org/10.2210/pdb6EXN/pdb, and 7B9V. The data supporting the findings of this study are available from the corresponding author upon reasonable request. Source data are provided with this paper.

## Code availability
Scripts for data pre-processing and downstream processing are available from github.com/luslab/psiclip (ref. [66]), https://doi.org/10.5281/zenodo.4439637.

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

## Acknowledgements

The authors dedicate this article to the memory of Kiyoshi Nagai (1949–2019) and Chris Oubridge (1966–2020). We thank Benjamin Porebski for sequencing the cDNA libraries. We also thank Clemens Plaschka and Flora Lee for critical reading of the manuscript. This work was supported by funding from the Medical Research Council (MC_U105184330), the European Research Council under the European Union's Seventh Framework Programme (FP7/2007-2013)/ERC grant agreement (617837). L.M.S. was supported by a Boehringer Ingelheim Fonds Fellowship. This work was supported by the Francis Crick Institute which receives its core funding from Cancer Research UK (FC010110), the UK Medical Research Council (FC010110), and the Wellcome Trust (FC010110). N.M.L. is a Winton Group Leader in recognition of the Winton Charitable Foundation's support towards the establishment of the Francis Crick Institute. N.M.L. and J.U. are additionally funded by a Wellcome Trust Joint Investigator Award (103760/Z/14/Z), and N.M.L. receives funding from the MRC eMedLab Medical Bioinformatics Infrastructure Award (MR/L016311/1) and core funding from the Okinawa Institute of Science and Technology Graduate University.

## Author contributions

L.M.S. established the method psiCLIP, with advice from M.H. and J.U. L.M.S. and C.C. designed psiCLIP experiments for SmB, Prp16, and Prp22. L.M.S. generated yeast strains, purified the helicase mutant proteins, and performed the psiCLIP experiments. L.M.S. and A.J.N. prepared the splicing substrates and performed the biochemical experiments. A.J.N. cloned, expressed, and purified Prp18. S.M.F., A.J.N., and C.O. provided experimental advice and contributed to data interpretation. C.C. performed all computational analysis. C.N. supported the generation of yeast strains and optimized debranching steps. The manuscript was written by L.M.S. and C.C., with input from all authors. K.N., J.U., and N.M.L. initiated and coordinated the project.

## Competing interests

The authors declare no competing interests.
