## [Peer Review File · Nature Communications]

REVIEWER COMMENTS

Reviewer expertise

Reviewer #1: RNA-seq analysis, splicing

Reviewer #2: spliceosome, structure, biochemistry

Reviewer #3: spliceosome, biochemistry

Reviewer #1 (Remarks to the Author):

Strittmatter and Capitanich et al. report their development of purified spliceosome iCLIP. Using this variation of iCLIP they investigate two spliceosomal helicases, Prp16 and Prp22, which are involved in spliceosome remodelling. The authors describe the binding profiles of both helicases on two substrates and at the different spliceosomal reaction steps they act on. Their work demonstrates that Prp16's binding to the substrate can adapt to accommodate intronic regions of various lengths. This supports the previous literature on Prp16 and strengthens the evidence that this helicase can act from a distance on the spliceosome's active site. Concomitantly, the authors' investigation of Prp22 illustrates how the 3' exon is not necessarily stably docked just before the second catalytic step (C* complex) but rather can be found bound to Prp22. This adds further insights into how Prp22 may proof-read splicing reactions.

These results are of interest and help us to complement and fill in gaps in our understanding of these important spliceosomal remodellers. Previous studies using smFRET or cryo-EM could locate both helicases to the periphery of the spliceosome, which removes them from the catalytic core they are thought to remodel, and suggested that they can act from a distance. The reported data in the submitted article confirm this model and further illustrate the dynamic character of the spliceosome, which have rendered it a difficult subject to study.

The manuscript is written in a clear and concise fashion and the presented data support the authors' claims.

Major concerns:

While I have to state that I am no expert in iCLIP and associated methodologies, the general analysis and interpretation of the data appears to be sound and reasonable.

I nevertheless want to voice some concerns about data presentation/interpretation. As the authors themselves point out in the discussion "absolute crosslinking levels in psiCLIP are affected by the variable crosslinking efficiencies of nucleotides and amino acids that can lead to biases at the sequence level" and "[that this] is important to take into consideration when making comparisons between transcripts and/or proteins".

I also observe that throughout the paper the number of crosslinks reported, overall as well as per position of interest, can vary strongly between replicates. I take this as an intrinsic aspect of iCLIP data and understand that many researchers utilising genome wide CLIP experiments report the summed replicates. But in such cases the represented profile can be dominated by a replicate that reported much more crosslinking than others. In this instance, this may also in particular influence the representation and interpretation of the results reported on Prp16 binding to ACT1 and UBC4.

Hence, for text and figures associated with p. 12 and figure 3c-d): Can the authors exclude these contributions to their conclusion that different positions are preferred on each transcript (20 nt downstream of the BP for UBC4 and 30 nt for ACT1, respectively)? A look at the replicates in figure S3 gives me the impression that regardless of the used transcript Prp16 interacts with the 40-50 nt window upstream of the branchpoint. This can be purely intronic (ACT1) or also cover the second exon

(UBC4).

In the following experiments it is then shown that Prp16 binding can adapt to cover the full space between BP and 3'SS. How do the authors see the observed Prp16 binding to the second exon of UBC4 in this context?

In addition, can Prp22 binding to the start of the second exon really quantitatively be described as "dramatically decreased" between complex C* and P (p.15 and figure 5)? The data is from two independent experiments and intronic Prp22 binding in complex C* may bleed together with Prp22 binding to the second exon.

Minor comments:

Pertaining to Figure 1 and manuscript page 8:

It would be helpful for the reader's orientation and navigation of the data if in figure 1 panel g) the same region of interest was displayed (e.g. all aligned at the 3' boundary by the Sm binding site). In addition, the U1 snRNA data and its binding profile are used to draw a generalised binding model for all snRNPs. It would therefore also be helpful if the indicated region (purple boundaries) could also be displayed on the zoom-ins for the other snRNAs.

While the crosslinking pattern immediately upstream of the SmB binding site (U1 snRNA A545 to U556) is easily discernible and appears to occur for all snRNPs, it is harder to judge if the further upstream crosslinking at U1 snRNA G535 and A536 may be a generalisable feature for the other snRNPs. Though, I concede that I may have overinterpreted the author's statement on this point.

Regarding the material and methods: I cannot find any mention on how much splicing substrate was added to the splicing reaction. How is this in relation to any RNA present in the splicing extract?

The same comment also applies for recombinant Prp18. How much is added to the Prp18 depleted splicing extract? I presume this correlates with endogenous Prp18 levels. (Is concentration having an influence – or it doesn't matter for what we're looking at in this paper?)

Additional suggestions:

- Figure 4 d) cartoon of splicing substrate could be made to include stem loop.
- Also pertaining to Figure 4h) and i): The introduced stem loop encompasses an additional 40 nucleotides. It would be nice for the reader if the zoom-in window shown in i) would encompass the full stem loop.
- It may be helpful to the reader if cartoons for mRNA representation, as seen in the main figures displaying the averaged psiCLIP data, were also displayed underneath the complimentary supplementary figures.
- Figure 6a-c) and Figure S6 d-f): panels should occur in the same order.

Typos:

- P. 9 Figure 2 (legend) "The U2 snRNA region highlighted in purple is shown in G in the crystal structure." Should be "shown in h", I presume.
- P. 14 Care should be taken with referencing the correct figure panels within this portion of the manuscript (several wrong figure (panels) referenced on this page).
- P. 14 2nd paragraph stem-loop instead of stem loop.
- P. 15 mentioning of Figures 5b in the text got lost.
- Figure S5b Prp2 instead of Prp22 (P complex plot)

Reviewer #2 (Remarks to the Author):

Strittmatter et al. have devised an extension of the conventional iCLIP procedures, by applying UV-crosslinking technology to yeast spliceosomes assembled in vitro. Well defined spliceosome stages are prepared by genetic depletion, substrate mutations, and/or add-back of mutated dominant negative helicases. RNA crosslinks to proteins are then isolated through pull-down of the tagged protein and subsequently mapped by high-throughput reverse-transcription assays using standard iCLIP procedures. The authors demonstrate the feasibility of the method by analysing RNA crosslinks to Prp16 and Prp22 RNA helicases. For Prp16, which functions in the conversion of the C to the C* complex, it is found that the binding window of the protein extends over the region between the branchpoint and the 3' splice site of the pre-mRNA. Enlarging the region to ~60 or ~80 nucleotides by using mutant substrates likewise leads to an extension of the binding window and an artificial stem-loop is not touched by the Prp16 protein. For Prp22, involved in the transition between the C* and P complex (step 2 of splicing), the authors found that the protein shifts from a position close to the 3' splice site in C*, to a downstream exonic site in the P complex. Finally, psiCLIP could be used for analysing the interaction of two splicing factors, as the authors demonstrate for Prp18 and Prp22 in the C* to P transition. Using a pre-mRNA with a mutated 3' splice site, they found that wild-type Prp22 does not bind the substrate 3' splice site, whereas an ATPase mutant of Prp22 still does.

This work demonstrates that the extension of iCLIP methodology to yeast spliceosomes prepared in vitro is feasible and allows an RNA-protein crosslinking analysis of purified spliceosomal assembly intermediates. The manuscript is at times difficult to understand and suffers from major shortcomings relating to presentation, and methodology and data analysis. Furthermore, it does not provide functional advance on our understanding of splicing and the role(s) of RNA helicases. No attempt is made to use the data to develop a working model beyond what is already known. Because of these shortcomings I cannot recommend publication of the manuscript in Nature Communications.

Major concerns:

On p. 4, the method is explained in detail and Figure 1a is referenced. The text refers to the C-to-C*-to-P transition(s) and the first part of the flow scheme is not mentioned. However, the SmB pulldowns are used in Figure 2, and no reference is made to Figure 1a. It is stated that the SmB pulldowns should serve as the first test of the new psiCLIP technique. Unfortunately, this first test is not documented in detail. In their experiments, the authors use a mixture of complexes ranging from early to first-step spliceosomes (p. 7) to analyse crosslinks to the U snRNAs and the two pre-mRNAs. This approach is problematic, as the crosslinks detected cannot be attributed to one particular spliceosome, and any dynamic component in the protein-RNA interactions between the complexes is lost. The authors also fail to perform a thorough analysis of their data. Here are a few questions that beg for an answer: What is the crosslinking efficiency? How many crosslinks can be correlated with proximities in the structures, and how is such proximity correlated with the number of crosslinks at a particular nucleotide? Can the non U crosslinks be rationalised from the structural data? How many false positives are detected? Considerations of this kind, backed up by statistical analysis, would have to be incorporated in any test of a new methodology, but they are missing in the present manuscript.

Figure S3 clearly shows variations between the three replicas. It looks as if there is a difference in the total number of crosslinks, and that there is a difference in the distribution of the crosslinks across the RNA. The authors give no explanation for these variations (or a statistical argument for why they can be ignored). This is troublesome, as later only two replicas are presented (Figure S4, S5), with the claim that the data are consistent across the replicates. I think that a measure of this consistency is needed.

The data analysis and presentation is at some points misleading. The authors devote some space to

transforming the primary data into Gaussian-smoothed curves, which are then plotted. This raises two major questions. First, what is the reason for presenting the data in this way, and why do the authors use different window sizes in the various figures for this mathematical transformation? I think bar graphs would be closer to the real data. This is most apparent in Figure 4i, where there are more -FLAG than +FLAG crosslinks upstream of the branchpoint (stem-loop panel). This clear difference is unfortunately not visible in Figure 4h, where data are normalised and Gaussian-smoothed. Second, the use of different Y axis scales is simply misleading. In Figure 4, for example, when one compares panels e and f, it is clear that the difference is the peak at ~30, which is present only in the WT; otherwise the profiles of the WT and 20nt extension are almost identical. This is readily apparent when a horizontal line is drawn at $y=0.025$. The same holds for panels g and h, with only a dip in numbers before the stem-loop in panel h. A plot of the difference would be more revealing.

Figure 3: The transient binding of Prp16 to the C complex was investigated by Prp16-psiCLIP on three different C complex preparations. However, the authors make no mention of any attempt to determine the purity of these complexes. This would be important, in particular as the AC substrate is expected to generate significant amounts of C* complex.

Figure 4: Prp16-psiCLIP was used to answer explicitly the question of how Prp16 acts from a distance downstream of the 3' splice site to pull the U2 branched structure out of the catalytic centre. However, the experiment fails to capture time-resolved dynamics of Prp16 functions.

These experiments cannot distinguish between simple binding/dissociation/rebinding events and the possibility that Prp16 moves in a 3'-to-5' direction on the pre-mRNA – a major question that the authors had initially set out to answer. Thus, compared with previous biochemical and structural data, we do not learn anything new about the mechanism of action of Prp16.

Nevertheless, the authors make the interesting observation that Prp16 shows comparatively few crosslinks to exon 2 sequences, irrespective of the variation in length of the intron between the branchsite and the 3' splice site. Even so, the authors do not discuss the reason for this apparent discrimination against exon 2 sequences. This would have been desirable, as it stands in complete contrast to the behaviour of Prp22.

In Figure 6, the peak of Prp22-K512A in panel c, marked with an asterisk, has an equivalent peak in panels a and b, at a similar position and with a similar intensity (related to the Y-axis scaling problem; see above, "The data analysis..."). However, in the text it is stated that the peak of Prp22-K512A in the second exon is enriched in the canonical substrate (panel c) as compared with the dG substrate (panel b). It is difficult to follow the authors' argument in the discussion (p. 20) when they say that their psiCLIP data on Prp22 and Prp18 depletion support the proofreading mechanisms proposed before. To establish a causal relationship would require demonstration that the dG substrate is in fact an erroneous substrate that is eliminated by proof reading.

Moreover, if I understand the experiments correctly, the authors' conclusions about apparent proofreading effects was based simply on mRNA reads, and had nothing to do with their psiCLIP approach. Thus, they simply confirm what has already been shown by others.

Minor points:

One of the authors (SMF, p. 1) is actually thanked in the Acknowledgements for reagents (p. 21).

Textual: some sentences are difficult to comprehend. For example, p. 4, 1st paragraph of Results. While "conventional iCLIP" is explained, it is not clear what "the method" is that was adapted. On p. 14, bottom, as a conclusion, the authors state "[...] that psiCLIP has a large dynamic range to test hypotheses about helicase remodelling [...]". There is little meaning to these words: what is the large dynamic range and how is it related to hypotheses-testing? On p. 16: the "Cryo-EM structure of [...] shows the helicase in green [...]". On p. 3, last paragraph: what do the authors mean by "global rules of splicing"? While this is a catchy phrase, it is meaningless without a precise description.

It is essential to quote the PDB ids of the structures shown in the figures (Figures 2h, 3e, and 5a,b).

Figure 1e: the boxes on the blots are counterproductive, as they introduce a visual bias for the reader. What are the numbers associated with the quantification?

On p. 7 it should be mentioned that the Sm proteins have also been modelled in crystal structures (not cryo-EM; that is, at least the listed references describe crystal structures)

Which pre-mRNA was used in Figure 2e?

Figure 3c: The difference between AC and AG symbols can only be guessed at.

Figures 4 e-h: Are the data quantitatively comparable in the horizontal axis? In other words, are the total numbers of crosslinks similar in the blue areas? Looking at Figure 4i, I would presume not. However, the authors do not address this point. An analogous point can be made for Figures 3c and d. Therefore I consider that a thorough re analysis of the data is essential.

Figure 4i: The zero on the X-axis should be a ten.

Figure 5d: Incomplete labels (cf 3a).

p. 15, bottom: The number 0.0003% looks strange, as it suggests an accuracy that I do not believe can be achieved experimentally. How many reads lie behind this number?

p. 16, line 3: How was the RNA "predicted"?

p. 20, centre: What would constitute an "incompletely assembled spliceosome"?

p. 21: "N.M.L"

p. 26, bottom: Tarasov must be cited for Sambamba, and not for the strandedness of psiCLIP. Also, further down, what were the reads mapping to, if not to the custom transcriptome? The idea of the "internal spike" is interesting, but more details need to be supplied.

Reviewer #3 (Remarks to the Author):

Pre-mRNA splicing is an extremely complex and highly dynamic process that takes place in a multi-megadalton RNA-protein complex, the spliceosome. The spliceosome assembles in a highly ordered series of steps that involve the addition of many factors, structural rearrangements, two catalytic reactions, and finally release of the spliced RNA and dissociation of the complex. This manuscript describes the development and application of a sophisticated biochemical approach referred to as psiCLIP to pin point the position within RNAs that proteins bind at different stages in the splicing cycle. This technically challenging approach involves assembling spliceosomes in vitro, UV irradiation to covalently crosslink proteins to the RNAs in the complex, recovery of the crosslinked protein-RNA adducts, followed by cDNA library preparation, high-throughput sequencing, and bioinformatic analysis of the data. Although a similar approach has been used to analyse protein-RNA interaction in live cells, the protocol had to be adapted for this novel in vitro application. This involved trials to titrate the UV treatment, RNA digestion and various reagents, and included meticulous use of controls. Eight RNA-stimulated ATPases are implicated in promoting structural changes and as fidelity factors for quality control of the process. The work here focuses on two of these, Prp16 and Prp22. Following validation of the procedure using the well characterised SmB protein, interactions of the Prp16 protein prior to the second catalytic reaction were mapped by stalling spliceosomes at complex C, using a

mutant form of Prp16 and with two different pre-mRNAs with canonical or mutated 3' splice sites. Crosslinks for both wild-type and mutant Prp16 were mapped throughout the region downstream of the intron branch site as far as the 3' splice site, even when the length of this region was extended by 40 nucleotides. Moreover, the finding that a stem-loop structure inserted in this region did not prevent splicing is thought to indicate that Prp16 does not need to translocate through this region.

Analogous experiments were performed to map Prp22 interactions in complex C*, immediately prior to exon ligation, and in complex P immediately after the second trans-esterification reaction. Prp22 appears to bind a wide region from the branchpoint downstream into the second exon in complex C* but with more localised binding in the second exon in complex P. Most interestingly, depletion of the second step factor, Prp18, resulted in loss of WT Prp22 binding but not of the mutant Prp22 binding. This is interpreted to indicate that binding by WT Prp22 is destabilised upon Prp18 depletion, which is compatible with structural studies.

In summary, this is a sophisticated biochemical analysis, with interesting results that confirm, extend and clarify previous observations and which will be valuable to the splicing community. From a methodological point of view, the ability to compare protein-RNA interactions in successive splicing complexes offers a more dynamic view of these interactions than was previously possible and it would be informative to extend this approach to other splicing factors. The work seem to have been performed to a very high standard and, for the most part, the authors have carefully considered alternative interpretations of their data and appropriately qualified their conclusions. I suggest below some minor considerations:

1) as Prp22 was observed to interact with the second exon just downstream of the 3' splice site, it would seem appropriate to discuss how this relates to binding of other proteins, e.g. Prp8 and Prp18, in the same region.

2) On p19/20 in the Discussion, binding of Prp16 or Prp22 over extensive regions of RNA is interpreted as reflecting dynamic binding and dissociation or translocation of the proteins along the RNA molecules. As the data come from populations of splicing complexes, widespread cross-links could reflect heterogenous binding among distinct complexes in the population rather than movement within individual complexes, although it might indeed also reflect snapshots of translocating or repeatedly binding Prp22 on different pre-mRNAs. This should be clarified.

3) bottom of page 19, there is a confusing sentence: "In complex P, Prp22 binding shifts to a narrower region in 3' exon, and this exonic binding pattern is similar between complex C* and complex P."

4) Fig 6 and on p20, "Prp18 depletion abrogates WT Prp22 binding..." This interesting result might be explained by an alternative (hypothetical) role for Prp18, modulating the ATPase activity of Prp22. In this scenario, the absence of Prp18 might cause the ATPase of Prp22 to be over-active, resulting in rejection of spliceosomes at that stage, whereas the mutant Prp22 with reduced ATPase activity bypasses this proof-reading step and bypasses the need for Prp18. Has the effect of Prp18 on Prp22's ATPase activity been tested?

5) A minor point: in Fig 5d the results for the C, C* and P complexes are plotted in the opposite order to the legend and to the order of splicing complex progression, which could be confusing.

Jean Beggs

PsiCLIP reveals dynamic RNA binding by DEAH-box helicases before and after exon ligation

Response to reviewers

Reviewer #1:

Strittmatter and Capitanich et al. report their development of purified spliceosome iCLIP. Using this variation of iCLIP they investigate two spliceosomal helicases, Prp16 and Prp22, which are involved in spliceosome remodelling. The authors describe the binding profiles of both helicases on two substrates and at the different spliceosomal reaction steps they act on. Their work demonstrates that Prp16's binding to the substrate can adapt to accommodate intronic regions of various lengths. This supports the previous literature on Prp16 and strengthens the evidence that this helicase can act from a distance on the spliceosome's active site. Concomitantly, the authors' investigation of Prp22 illustrates how the 3' exon is not necessarily stably docked just before the second catalytic step (C* complex) but rather can be found bound to Prp22. This adds further insights into how Prp22 may proof-read splicing reactions.

These results are of interest and help us to complement and fill in gaps in our understanding of these important spliceosomal remodellers. Previous studies using smFRET or cryo-EM could locate both helicases to the periphery of the spliceosome, which removes them from the catalytic core they are thought to remodel, and suggested that they can act from a distance. The reported data in the submitted article confirm this model and further illustrate the dynamic character of the spliceosome, which have rendered it a difficult subject to study.

The manuscript is written in a clear and concise fashion and the presented data support the authors' claims.

1.1 Author's response:

We would like to thank reviewer 1 for their interest in our manuscript and recognition that our work fills gaps in a very difficult subject - that of the dynamic character of the spliceosomal helicases. We now offer the following responses to their specific comments below.

Reviewer 1 Major concerns:

While I have to state that I am no expert in iCLIP and associated methodologies, the general analysis and interpretation of the data appears to be sound and reasonable.

I nevertheless want to voice some concerns about data presentation/interpretation. As the authors themselves point out in the discussion "absolute crosslinking levels in psiCLIP are affected by the variable crosslinking efficiencies of nucleotides and amino acids that can lead to biases at the sequence level" and "[that this] is important to take into consideration when making comparisons between transcripts and/or proteins".

I also observe that throughout the paper the number of crosslinks reported, overall as well as per position of interest, can vary strongly between replicates. I take this as an intrinsic aspect of iCLIP data and understand that many researchers utilising genome wide CLIP experiments report the summed replicates. But in such cases the represented profile can be dominated by a replicate that reported

much more crosslinking than others. In this instance, this may also in particular influence the representation and interpretation of the results reported on Prp16 binding to ACT1 and UBC4.

Hence, for text and figures associated with p. 12 and figure 3c-d): Can the authors exclude these contributions to their conclusion that different positions are preferred on each transcript (20 nt downstream of the BP for UBC4 and 30 nt for ACT1, respectively)? A look at the replicates in figure S3 gives me the impression that regardless of the used transcript Prp16 interacts with the 40-50 nt window upstream of the branchpoint. This can be purely intronic (ACT1) or also cover the second exon (UBC4).

1.2 Author's response:

Firstly, many thanks for this nuanced comment. The overall number of crosslinks does differ between experiments which is an intrinsic aspect of iCLIP, but the main source of variation is the RBP studied, which is due to different crosslinking efficiency of each RBP. In addition, there is variation between replicates of data presented for Prp16 in figures 3c,d (also in Figure S3), as noted both by reviewers 1 and 2. This is because Prp16 very weakly crosslinks to RNA (also compared to Prp22), which leads to data with low coverage. Producing highly specific iCLIP libraries for such low-efficiency crosslinkers is extremely challenging, due to increased possibility for contamination from high-efficiency crosslinkers - therefore, we made all the effort to maximise specificity by highly stringent multi-step purification, but such high specificity comes at a price of lower coverage. We therefore scaled up the experiment to obtain as much coverage as possible. Nevertheless, it is well known that in CLIP, just like in an RNA-seq, experiments with lower coverage replicates will result in more variation - which can be seen when comparing Prp16 with Prp22 (where we obtained higher coverage). This is why we opted for preparing the larger numbers of replicates for Prp16 and summing them up in the main figures. We performed 3 replicates in 3 different, but ultimately similar conditions, essentially amounting to 9 replicates in total that are presented in Figure S4. Across two different splicing substrates, each with an untagged control condition, this amounts in total to 36 CLIP experiments, which is unheard of in the CLIP field - an effort that we believe can ameliorate the increased variability of Prp16 data. We provide the replicate data separately in the supplementary information alongside the total number of crosslinks for each experiment. Actually experiments with fewer crosslinks are less trustworthy and so summing the replicates is the equivalent of taking a weighted average. We purposefully want to bias towards experiments with higher coverage as these are more reliable. Although, as you can see from the replicates individually the binding pattern itself is reproducible, even where the intensity is more variable.

To address your specific question “**Can the authors exclude these contributions to their conclusion that different positions are preferred on each transcript (20 nt downstream of the BP for UBC4 and 30 nt for ACT1, respectively)**”, we include below some additional comparative analysis:

If we consider all data, presented below for *UBC4* in blue and *ACT1* in red, with a 20nt smoothing window, the crosslinking efficiency on the different substrates differs, which makes it difficult to determine if Prp16 crosslinks to the same or different region on the two splicing substrates. Below we show tagged (solid lines) and no tag controls (dashed lines) separately.

In order to make the data more comparable and to assess if peaks are shifted, we subtracted the no tag control from each tagged sample and subsequently normalised the data to the maximum peak in each sample within this window (see figure below).

Here, we see that most of the time Prp16 seems to be crosslinking further downstream on the *ACT1* substrate (red). In addition, we performed a t-test to ask whether the positions of the height of the peaks in this graph are significantly different with $\alpha=0.05$, which they are with $p<0.001$. We performed this for the raw counts too, and we also find a significant difference in position (see table below). In summary, this analysis shows clearly that the distributions are indeed different, meaning that Prp16 crosslinks to different positions in the two different substrates. While this indicates a difference in binding profile between the two substrates, we add a note of caution in the paper, explaining that crosslinking profiles may not directly represent the binding profiles of RBPs due to technical differences, such as for example the nucleotide preferences of crosslinking. This uncertainty

was another reason to perform psiCLIP experiments with unnaturally extended versions of the *ACT1* transcripts, which enabled an alternative approach to compare different transcripts.

CTRL MINUS NORM			
	average position	standard deviation	t-test p value
ACT1	27	4	2.53959E-05
UBC4	19	3	

RAW COUNTS			
	average position	standard deviation	t-test p value
ACT1	29	11	0.006832003
UBC4	16	4	

Author action:

We have clarified the section about the crosslinking peaks of Prp16 on *ACT1* and *UBC4* in “psiCLIP reveals broad binding of Prp16 downstream of brA” and pointed out in the text the caveats of concluding binding positions from crosslinking position.

We have included this analysis of difference in binding position as Figure S4h, and refer to this in the text.

“... On both splicing substrates, the binding pattern of Prp16 is widely spread, producing prominent crosslinks up to 40nt downstream of the brA and extending well beyond the predicted 9nt for the occluded site of a DEAD/H-box helicase. On *ACT1* Prp16 shows a prominent peak around 30 nucleotides downstream of the brA, whereas on *UBC4* the main peak occurs 20 nucleotides downstream of the brA, though binding spans the entire region between the brA and the second exon for both substrates (Figure 3e,f – sum of all replicates, Figure S4c– three replicates shown separately). Although the position of the main crosslinking peak is significantly different between the two splicing substrates ($p < 0.001$, Student’s t-test, Figure S4h, Supplementary Table 1), comparisons between substrates need to be interpreted with caution, as crosslinking efficiency may be affected by the underlying RNA sequence ...”

In the following experiments it is then shown that Prp16 binding can adapt to cover the full space between BP and 3’SS. How do the authors see the observed Prp16 binding to the second exon of *UBC4* in this context?

1.3 Author’s response:

We thank the reviewer for this comment, as by addressing it we could now discuss further insights. As described in section “Prp16 binding depends on the brA to 3’SS distance” our data describes that Prp16 binding extends from the brA up to +40nt for both *UBC4* and *ACT1*. Mayer et al. 2011 (DOI: 10.1016/j.molcel.2011.07.030) predicts most unstructured yeast introns having their 3’SS within 45nt of the brA, so that Prp16 would bind between the brA and the 3’SS. *UBC4* has a short distance of only 26nt so that Prp16 binding extensively “spills over” into the exon. Even on *ACT1*, which has a brA-3’SS distance of 44nt, some crosslinks can be found in the second exon meaning that exonic binding is not actively disabled (compare figure of raw crosslinking data below). Further, Mayer et al. 2011 have found that long natural introns fold into a secondary structure to achieve an effective

brA-3'SS distance of ~45nt. For randomised intron sequences, the authors found that the effective distance is increased to ~60nt.

Taken all these findings together, we don't expect Prp16 to actively discriminate against the exonic sequence, but that Prp16 binding is constrained to a region up to ~40nt downstream of the brA by other spliceosomal components and this distance correlates with the (effective) distance between brA and 3'SS to make splicing most efficient. When providing single stranded regions such as through the randomised extension of *ACT1* (compare Figure 4), the helicase is offered a more extended region for binding - this shows how the helicase accommodates the splicing of transcripts with various distances.

Raw crosslinks for Prp16 shown relative to branch A (0), the second dashed line marks the 3'SS:

Author action:

To reflect more the nuance of this binding we have edited our text both in the results and discussion section:

“... Importantly, Prp16 binding did not appear strictly constrained by the 3'SS, as binding could be detected in the second exon for UBC4. [...] Thus, our results indicate that Prp16 binds downstream of the brA to the entire length of accessible single-stranded RNA, and does not strictly discriminate between introns and exons ...”

“... Instead of binding at a fixed distance from the brA, we found that Prp16 binds over the full available single-stranded RNA region between the brA and 3'SS, even when this distance is extended with

unstructured RNA. Nonetheless, most of the Prp16 psiCLIP signal occurred within ~45 nucleotides downstream of the brA, which correlates with the most common distance between brA and 3'SS for unstructured yeast introns ...”

In addition, can Prp22 binding to the start of the second exon really quantitatively be described as “dramatically decreased” between complex C* and P (p.15 and figure 5)? The data is from two independent experiments and intronic Prp22 binding in complex C* may bleed together with Prp22 binding to the second exon.

1.4 Author’s response:

We agree with this view after careful re-examination of the data, and we now comment only on the change in intronic binding and the constant exonic binding.

Author action:

We have changed the text to reflect this:

“... Importantly, the major psiCLIP peak observed for Prp22 in complex P on the second exon is also observed in complex C* (Figure 5e-h), indicating that Prp22 binds at similar positions within the second exon both before and after exon ligation ...”

Reviewer 1 Minor concerns:

Pertaining to Figure 1 and manuscript page 8:

It would be helpful for the reader’s orientation and navigation of the data if in figure 1 panel g) the same region of interest was displayed (e.g. all aligned at the 3’ boundary by the Sm binding site).

Author action:

We have changed the figure (now Figure S2) accordingly to make it easier for the reader. Please note that this figure has been reduced in the main text, and the full figure is shown in the supplementary. From other comments it is clear that the way the manuscript was originally organised put too much spotlight on the SmB data, which was meant just for validation.

In addition, the U1 snRNA data and its binding profile are used to draw a generalised binding model for all snRNPs. It would therefore also be helpful if the indicated region (purple boundaries) could also be displayed on the zoom-ins for the other snRNAs.

1.5 Author’s response:

While, SmB might bind very similar with its core region as implied by our psiCLIP data, its flexible parts may bind differently in the different snRNPs. Unfortunately, we can only speculate here as the resolution within the cryoEM structures is not sufficient to model the distinct RNA nucleotides into density. The U1 snRNP of the pre-B complex is the only one we show in the manuscript, because it contains the only yeast Sm-ring structure within a spliceosome with high enough resolution to allow cross-validation of our data.

Author action:

To clarify this we have changed the text:

“... The 5' end of U1 snRNA extends into the direction of the SmB protein, thus explaining why psiCLIP detected crosslinks mainly at the beginning of the Sm site motif. [...] The psiCLIP data suggest that SmB has a similar configuration in the other snRNPs (Figure S2b-g)...”

While the crosslinking pattern immediately upstream of the SmB binding site (U1 snRNA A545 to U556) is easily discernible and appears to occur for all snRNPs, it is harder to judge if the further upstream crosslinking at U1 snRNA G535 and A536 may be a generalisable feature for the other snRNPs. Though, I concede that I may have overinterpreted the author's statement on this point.

1.6 Author's response:

As described above, we have changed the sections to emphasise more the similar crosslinking pattern for the core region of SmB. We think the suggested data presentation aligning the zoom-in onto the Sm-site strongly helps the reader to appreciate the similar binding positions around and in the Sm site and even indicates that the flexible region of SmB might well be oriented similarly in other snRNPs.

Regarding the material and methods: I cannot find any mention on how much splicing substrate was added to the splicing reaction. How is this in relation to any RNA present in the splicing extract?

1.7 Author's response:

The amount of RNA used in an in-vitro splicing reaction correlates with the amount of splicing components, such as snRNPs and other splicing factors, in the extract. In general, the amount of splicing substrate is titrated up until the splicing efficiency decreases, which implies that not enough splicing components are available to splice the added RNA effectively. In summary, reactions were performed as typical in the splicing field (described by Lin et al., 1985 (<https://pubmed.ncbi.nlm.nih.gov/2997224/>) and Umen & Guthrie, 1995 (<https://pubmed.ncbi.nlm.nih.gov/8548652/>))

Author action:

We have included into the methods section that a concentration of 2.5 nM of splicing substrate was used for *in vitro* splicing reactions.

The same comment also applies for recombinant Prp18. How much is added to the Prp18 depleted splicing extract? I presume this correlates with endogenous Prp18 levels. (Is concentration having an influence – or it doesn't matter for what we're looking at in this paper?)

1.8 Author's response:

Recombinantly produced protein is expected to also contain incorrectly folded protein. To achieve an amount of “active” Prp18 that is equivalent to the endogenous amount, a functional titration assay was performed. The lowest concentration was used that rescued the second step splicing defect of the depleted extract. We apologize that we had initially only included a total amount without a concentration.

Author action:

We have adjusted the methods section to include that Prp18 was added to reach a total concentration of 165 ng/mL in the splicing reaction for phenotype rescue.

Additional suggestions:

- Figure 4 d) cartoon of splicing substrate could be made to include stem loop.
- Also pertaining to Figure 4h) and i): The introduced stem loop encompasses an additional 40 nucleotides. It would be nice for the reader if the zoom-in window shown in i) would encompass the full stem loop.

Author action:

We have changed Figure 4 accordingly

- It may be helpful to the reader if cartoons for mRNA representation, as seen in the main figures displaying the averaged psiCLIP data, were also displayed underneath the complementary supplementary figures.

Author action:

We have added cartoons for mRNA representation in all relevant figures.

- Figure 6a-c) and Figure S6 d-f): panels should occur in the same order.

Author action:

We have changed Figure S6, so that the panels occur in the same order.

Typos:

- P. 9 Figure 2 (legend) "The U2 snRNA region highlighted in purple is shown in G in the crystal structure." Should be "shown in h", I presume.
- P. 14 Care should be taken with referencing the correct figure panels within this portion of the manuscript (several wrong figure (panels) referenced on this page).
- P. 14 2nd paragraph stem-loop instead of stem loop.
- P. 15 mentioning of Figures 5b in the text got lost.
- Figure S5b Prp2 instead of Prp22 (P complex plot)

Author action:

We thank you for the detailed feedback, and we have addressed the typos in the revised version of the manuscript.

Reviewer #2:

Strittmatter et al. have devised an extension of the conventional iCLIP procedures, by applying UV-crosslinking technology to yeast spliceosomes assembled in vitro. Well defined spliceosome stages are prepared by genetic depletion, substrate mutations, and/or add-back of mutated dominant

negative helicases. RNA crosslinks to proteins are then isolated through pull-down of the tagged protein and subsequently mapped by high-throughput reverse -transcription assays using standard iCLIP procedures. The authors demonstrate the feasibility of the method by analysing RNA crosslinks to Prp16 and Prp22 RNA helicases. For Prp16, which functions in the conversion of the C to the C* complex, it is found that the binding window of the protein extends over the region between the branchpoint and the 3' splice site of the pre-mRNA. Enlarging the region to ~60 or ~80 nucleotides by using mutant substrates likewise leads to an extension of the binding window and an artificial stem-loop is not touched by the Prp16 protein. For Prp22, involved in the transition between the C* and P complex (step 2 of splicing), the authors found that the protein shifts from a position close to the 3' splice site in C*, to a downstream exonic site in the P complex. Finally, psiCLIP could be used for analysing the interaction of two splicing factors, as the authors demonstrate for Prp18 and Prp22 in the C* to P transition. Using a pre-mRNA with a mutated 3' splice site, they found that wild-type Prp22 does not bind the substrate 3' splice site, whereas an ATPase mutant of Prp22 still does.

This work demonstrates that the extension of iCLIP methodology to yeast spliceosomes prepared in vitro is feasible and allows an RNA–protein crosslinking analysis of purified spliceosomal assembly intermediates. The manuscript is at times difficult to understand and suffers from major shortcomings relating to presentation, and methodology and data analysis. Furthermore, it does not provide functional advance on our understanding of splicing and the role(s) of RNA helicases. No attempt is made to use the data to develop a working model beyond what is already known. Because of these shortcomings I cannot recommend publication of the manuscript in Nature Communications.

2.1 Author's response:

We would like to thank reviewer 2 for their close attention to our manuscript and comments which have improved our manuscript substantially. We appreciate that in this work we are treading new ground in establishing psiCLIP as a tool to gain insights into mechanisms of ATPases in their native complexes. This study bridges the fields of structural biology and transcriptomics that have been only rarely integrated, and which tend to use different terminologies, presentation and analysis approaches - thus, we had to solve not just technical, but also presentation challenges in order to make the study easily accessible and interpretable for both fields. Perhaps the functional advances didn't stand clearly enough in the initial submission, we have clarified this in the revision and sum it up here:

- CLIP data are not measuring individual RNA molecules, but identify the position of binding by identifying crosslinking events from a population of RNAs. Thus, insights from CLIP are relative, based on the following types of comparisons: comparison of cDNA counts across different complexes, to give insights into remodelling of complexes / comparisons across different proteins bound to the same RNA / and comparisons between different positions on the same RNA. Our study takes the field further by performing these comparisons on two helicases within purified RNPs at multiple defined states.
- Information on RNA binding sites and the mechanism of helicases is not accessible from traditional structural biology methods, and it is an essential knowledge required for the field to move further (as Kiyoshi stressed in many conversations, also in Fica and Nagai, 2019, DOI: 10.1038/nsmb.3463). PsiCLIP allowed us to gain direct insights, for the first time, into the RNA binding positions and remodelling of DEAH-box helicases.
- By validating and characterising psiCLIP, we present a method that can be easily adapted in the future to study remodelling of protein-RNA contacts in large, purified RNPs from defined functional states.
- Most previous experiments using traditional methods have suggested that DEAH-box ATPases bind in a defined window and act in a directional and potentially processive manner on their RNA substrate. Conversely, our data indicate that binding is likely much more

stochastic and may cover the entire single stranded region available on the substrate. Indeed, depending on the type of complex that is studied, we observe both defined peaks of binding, as well as more diffuse binding along the substrate, suggesting that DEAH-box helicases may act from multiple positions along the substrate, pointing to a more complex mode of DEAH-box helicase activity.

Finally, although we compare our data with findings from structural biology, it is not the aim of our study to reach the same level of resolution, but rather to provide insights that complement structural knowledge. We now explain more clearly these advantages and shortcomings of our study both in the revised manuscript and in response to the specific points below.

We thank the reviewer to suggest including a working model to strengthen the impact of the manuscript for the audience. Following their suggestion, based on our data and the improved analysis thereof, we summarise our findings in the model “substrate binding by spliceosomal helicases during the catalytic stage of pre-mRNA splicing” (included as Figure 7). Additionally, we proposed a working model for Prp22 proofreading (included as Figure S8) that is based both on the presented psiCLIP data supplemented with findings from recent cryoEM studies.

Reviewer 2 Major concerns:

On p. 4, the method is explained in detail and Figure 1a is referenced. The text refers to the C-to-C*-to-P transition(s) and the first part of the flow scheme is not mentioned. However, the SmB pulldowns are used in Figure 2, and no reference is made to Figure 1a.

2.2 Author's response:

We apologise for the confusion this has caused and have now changed the text and the order of the figures to optimise the flow and make the referencing clearer.

Author action:

We have changed the order of figures in the first result section and adjusted the text to improve flow and avoid confusion.

It is stated that the SmB pulldowns should serve as the first test of the new psiCLIP technique. Unfortunately, this first test is not documented in detail. In their experiments, the authors use a mixture of complexes ranging from early to first-step spliceosomes (p. 7) to analyse crosslinks to the U snRNAs and the two pre-mRNAs. This approach is problematic, as the crosslinks detected cannot be attributed to one particular spliceosome, and any dynamic component in the protein–RNA interactions between the complexes is lost.

2.3 Author's response:

We apologize that the design of the SmB experiment was not clear enough, and we have added clarification as described below.

Reviewer 2 is concerned that we analysed a mixture of different spliceosome states in the SmB psiCLIP experiment. However, we did not aim at this stage to produce step-specific spliceosomes or to study spliceosome dynamics, but rather to use SmB to validate and optimise the protocol to (a) specifically isolate the specific protein (SmB) from purified spliceosome, and to (b) recapitulate the expected Sm binding site. We achieved both, which is a powerful argument in favour of the high stringency and positional accuracy of our method.

Based on the reviewer's comment, we searched the literature to find evidence for remodelling of the Sm contacts during the splicing cycle. When comparing the PDB files for all yeast spliceosome structures, we could not find any changes in the protein contacts made by Sm proteins in all complexes, implying a rather static configuration of the SmB protein within the mixture of spliceosomes that we analysed. That being said, the resolution of bound RNA is not sufficient to rule out any small changes (see author's response 1.5) and we could not find any biochemical analysis that would explore the dynamics of SmB proteins. Therefore, we find SmB proteins appropriate as an initial test of psiCLIP's resolution, but not to study spliceosome dynamics or step-specific spliceosomes. We will appreciate further insights, if the reviewer is aware of any expected dynamics in regard to the SmB ring.

In later parts of the manuscript, when we start to investigate the mechanism of DEAH-box helicases Prp16 and Prp22, step-specific complexes were prepared as stated in the reviewer's introductory statement - "Well defined spliceosome stages are prepared".

Author Action:

As described in response 2.2., we changed the flow and order of the initial figures and the description of the SmB data. We hope it is clear now, that SmB psiCLIP was only used as a validation step for the method rather than to study dynamics of the SmB ring during splicing.

The authors also fail to perform a thorough analysis of their data. Here are a few questions that beg for an answer: What is the crosslinking efficiency?

2.4 Author's response:

As described above, CLIP only yields relative information - absolute crosslinking efficiency can't be defined from CLIP data alone, and it is not a standard to define it. That said, the efficiency of UV crosslinking is meant to be low to ensure that each individual RNA molecule contains only one or a few crosslinking events, rather than crosslinking to all the bound proteins simultaneously - that would be problematic, as the sites of protein-RNA crosslink terminate cDNA synthesis, and so it would bias cDNA truncations to the ends of RNA molecules and binding sites. We optimised our crosslinking conditions towards this aim - we achieved detectable signal in the protein-RNA gel, while ensuring that the whole binding sites are covered by stochastic distribution of crosslinks without any evidence of over-crosslinking, where one would expect a bias of reverse transcription truncations to the 3' end of the binding sites. From our data we see a good distribution of crosslinking across the intron and exon so we are confident we are not over-crosslinking. We achieved this by UV dose titration as described in figure 1e. We now describe the used UV dose for each different complex and target protein in the method section.

Author action:

We have included the specific UV doses used into the method section.

How many crosslinks can be correlated with proximities in the structures, and how is such proximity correlated with the number of crosslinks at a particular nucleotide?

2.5 Author's response:

Again, insights from CLIP are relative, and while the question posed is interesting, we are not aware of any past study that would provide a definitive answer to this question. Current knowledge of how exactly nucleotides crosslink to peptides and how distance and geometry are correlated to the crosslinking level for natural nucleobases is still quite limited.

Unfortunately, it is not straightforward for our study to address this question further. First, structural models of the spliceosomal DEAH-box helicases are static. Second, none of the published cryoEM spliceosome structures show sufficient resolution to determine the identity of the nucleotides that are bound to Prp16 or Prp22. High resolution crystal structures of DEAH-box helicases are available, however, here the helicases were bound to non-natural poly-U sequences and were crystallized in isolation rather than bound to a spliceosome. In fact, this limitation, and the fact that even within a defined spliceosomal state the helicases might transiently contact a broad RNA region, are two of our primary motivations for developing psiCLIP. This is the first time such detailed binding can be observed experimentally in the native context and in solution for an RNP complex. Moreover, the data itself suggests we have captured intrinsic dynamics of the helicases which cryo-EM (that has shortcomings for dynamic domains) may never be able to capture in higher resolution binding than this method.

Despite not being able to correlate crosslinking with proximity, we have included the constraints provided by cryoEM and previous biochemical experiments wherever we could, and we have found psiCLIP results to be consistent. For instance for SmB psiCLIP, that was solely set up to optimise and verify that psiCLIP captures protein binding on several RNA substrates of a single protein (see response 2.3), we found one spliceosome structure that showed high enough resolution for snRNA and Sm-ring in one snRNP for comparison with our data. As described in response 1.5, we cannot map SmB binding onto the remaining U2, U4 and U5 snRNA as structural information is not available.

In general, psiCLIP's consistency with orthogonal data, both cryoEM and FRET, argue in favour of the validity of our positional insights.

Author action:

Due to limited structural information and the described shortcomings in determining the correlation of crosslinking signal and proximity, unfortunately we cannot include a more quantitative analysis of crosslinking efficiency and accuracy for individual nucleotide positions. Future studies, beyond the scope of this paper, are needed to establish such a thorough quantitative framework for iCLIP experiments.

Can the non-U crosslinks be rationalised from the structural data? How many false positives are detected? Considerations of this kind, backed up by statistical analysis, would have to be incorporated in any test of a new methodology, but they are missing in the present manuscript.

2.6 Author's response:

Reviewer 2 asks for an analysis of false positive signal - we assume the reviewer refers to crosslinking events that derive from non-specific proteins or RNAs, i.e., those that do not represent direct contacts of the helicases of interest. Indeed, this is exactly the core aspect of the CLIP method design. So in the respect of optimising the specificity, our methodology is not new. CLIP includes the step to evaluate false-positive signal at the stage of visualising the SDS-PAGE-separated complexes. Thus, even before sequencing, we have undergone many rounds of recursive experiments that employed this visualization to optimise the method specificity. In this respect, we have followed well-established guidelines. We have used a no-tag control throughout the manuscript to account for false positive signal that can derive from proteins or RNA non-specifically binding to the antibody or beads. Because of the high stringency and well-optimised CLIP conditions we use, the no-tag control generates very little signal, and throughout the manuscript we frequently normalise to it, in the case where it does generate more signal, ie. in Figure 4g,h we are careful to draw attention to this.

In terms of providing a statistical analysis, allow us to draw a parallel - in ChIP-Seq analysis of chromatin-bound proteins the data is often presented as Log2 fold change of IP over input and a

statistical test might be used to call significantly enriched regions - this is necessary when you have data over an entire genome and must prioritise which peaks you take forward for further analysis. In this instance we study a handful of specific substrates and show the reader all of our data, so a statistical peak calling is unnecessary and just as subjective as making decisions by eye, because you select the parameters.

Moreover, a comparison between psiCLIP data for multiple proteins purified from the same input material provide further evidence of positional specificity of data - we were reassured to find that results for the two helicases are very different - which would not be the case for false positive signal that is expected to be same (as both helicases are purified in the same way from similar complexes). Since our substrates do not contain photoactivatable uridines we would expect to see crosslinks at all nucleotides, however with variable efficiency. How this depends also on the protein that is crosslinked remains incompletely understood, as described in the last paragraph of the discussion. Thus, we see no reason why the observed non-U crosslinks would not be genuine crosslinks. As described in 2.5, we would not necessarily expect all crosslinks to correlate with contacts observed in structures, which are static snapshots. By contrast our method captures solution binding, which is most likely dynamic and may involve regions that become flexible during splicing and are not necessarily observed in the structures as interacting with a given protein.

In sum, we follow the steps of quality analysis that control for potential false-positive signal according to the current state of the art in the CLIP field.

Figure S3 clearly shows variations between the three replicas. It looks as if there is a difference in the total number of crosslinks, and that there is a difference in the distribution of the crosslinks across the RNA. The authors give no explanation for these variations (or a statistical argument for why they can be ignored).

This is troublesome, as later only two replicas are presented (Figure S4, S5), with the claim that the data are consistent across the replicates. I think that a measure of this consistency is needed.

2.7 Author's response:

We do appreciate our Prp16 data is more variable than our Prp22 data, please see our response to reviewer 1 in Author's response section 1.2, in addition to our response below.

A measure of consistency is indeed a good suggestion, therefore we have done a correlation analysis for the Prp16 experiments shown in Figure 3 for both assembly on *UBC4* and *ACT1*.

We focused on the correlation in **positional information**, rather than intensity, as none of our conclusions about Prp16 rely on the intensity information as we find this to be very variable between samples, as you note. The region -50nt to +50nt around the branch A was taken, and raw crosslinks normalised to the maximum crosslink in this window, then the relative score at each position was plotted against all other Prp16 samples in a matrix. As well as providing another way of assessing the similarity visually, we also calculated a Pearson correlation coefficient. Of note is the greater degree of variation on the *ACT1* substrate compared to *UBC4*, particularly with the WT Prp16 on the AC substrate, where the pearson correlation between replicates is very poor - this is reflected by the fact that our libraries were very small for this experiment as can be observed in the total number of crosslinks shown in Figure S4. For this reason we have decided to remove this experiment from our main figure, and show it now only in supplementary data.

Overall, alongside additional analysis performed in response to reviewer 1 (Author's response 1.2), we feel this correlation analysis will help readers to understand the variability in our Prp16 data.

As a comparison, we see the correlation between Prp22 samples to be much stronger, which reflects the fact it is a much stronger crosslinker to RNA, which is why we only needed two replicates for these experiments (see Response 2.7, Figure 2 below)

Response 2.7, Figure 1 - Correlation analysis of Prp16 data.

Response 2.7, Figure 2 - Correlation analysis of Prp22 data.

Author action:

Both new supplementary correlation analysis, and statistical analysis of difference in binding positions for Prp16 on ACT1 and UBC4 have now been included in supplementary figure S4 as S4f,g and h. We have also removed WT Prp16 on the AC ACT1 substrate from the main figure, owing to the poor correlation between replicates we have now quantified.

We have changed the text in several places to make the variation in the data clearer to the reader:

“... Indeed, we found that Prp16 makes similar RNA contacts regardless of the stalling method, demonstrating the high reproducibility of the psiCLIP method (Figure 3e,f, Figure S4c,e). Nonetheless,

we found that poor crosslinking efficiency for Prp16, compared to Prp22 for example, resulted in increased stochastic variation in the crosslinking profiles between replicates (Figure S4b,d,f,g) ...”

Legend for Figure 3c,d:

“... Note that WT Prp16 binding on AC ACT1 substrate is excluded here due to small library sizes and low reproducibility (Figure S4f). ...”

The data analysis and presentation is at some points misleading. The authors devote some space to transforming the primary data into Gaussian-smoothed curves, which are then plotted. This raises two major questions. First, what is the reason for presenting the data in this way, and why do the authors use different window sizes in the various figures for this mathematical transformation? I think bar graphs would be closer to the real data.

2.8 Author's response:

We agree that the histograms represent the raw data more closely. Our rationale for performing the smoothing is aimed towards enabling the human eye to see the general binding pattern, as it is not easy for our eyes to see a pattern when overwhelmed with too many tiny bars at each nucleotide, as explained below:

1. RNA-binding proteins do not bind to single nucleotides, and the investigated helicases lack sequence specificity. Therefore, we are interested in binding patterns across larger regions, not at individual nucleotides. Even sequence-specific RBPs often crosslink quite variably across the motif, and sometimes actually outside of their binding motifs (compare Figure 5 in Feng et al., 2019, DOI: 10.1016/j.molcel.2019.02.002; <https://www.sciencedirect.com/science/article/pii/S1097276519300929#fig4>). The large variations in crosslinking efficiency between nucleotides make nucleotide-resolution data spiky at individual sites, even though the true binding is not as spiky - therefore the smoothing yields more informative binding profiles.
2. Very few studies outside of our lab have ever been showing nucleotide-resolution CLIP information at high resolution on individual genes. In fact, we are not aware of any paper that would have evaluated CLIP data comparatively at nucleotide resolution on narrow binding regions (<100nt). We and close colleagues have done it in the past on large regions >700nt, where there's no need for smoothing because nucleotide resolution is anyway lost due to zooming out over large regions. Most other labs actually don't even consider nucleotide resolution when analysing binding trends. They either compare counts per mRNA, and the labs producing vast amounts of data for the ENCODE consortium evaluate full reads to derive binding peaks (called Narrowpeaks), meaning they even ignore crosslinks. Unfortunately, we are not aware of any review that would sum up the visualisation approaches comparatively, and we just want to highlight that this manuscript is breaking new grounds by comparative visualisation of such small binding regions.
3. Since there are so many samples in the study, it is necessary that we compare them in a single plot in main figures; simply to make comparison easier for the reader. This, unfortunately, can only be done if we show each replicate as a line which requires some smoothing, otherwise the results will be presented with various distracting spikes and cannot be discerned by the human eye.

In sum: Showing crosslinking profiles at nucleotide resolution would highlight the absolute profiles, which will highlight crosslinking biases, while hiding the comparisons of more general binding profiles from the human eye. It is for this reason that the manuscript uses smoothing - as a visualisation tool

for comparative analysis, rather than trying to hide any crosslink biases. The field knows these biases well, and accepts that the shape of a crosslinking profile doesn't reflect the binding profiles in an absolute manner; we will stress this further in the text of the revised paper. Smoothed crosslink profiles, instead, allows our eyes to see the changes across conditions, which hold the mechanistic information. We have opted to use a 10 nt window for the Prp22 data which is in the same range as the occluded site of a DEAH-box helicase, which is predicted to be 9nt (He et al., 2017, DOI: 10.1261/ma.060954.117). We initially used a broader window for Prp16 because the data is more variable and so noisier to plot multiple replicates on the same graph, the broader window lets the reader more easily assess the conclusions we are making. However, we have now changed the smoothing for Prp16 in the revised manuscript to be 10nt for consistency.

To provide assurance to the reviewer that smoothing is not obscuring any vital information, we provide example plots below of Prp22 and Prp16 with 5nt, 10nt and 20nt Gaussian smoothing windows alongside the raw crosslinks to demonstrate that our choice is simply to visually highlight the main peaks, and that no data is lost that impacts the conclusions in the manuscript (**2.8 Author's response, Figure 1**). We also show an alternative smoothing method - a simple moving average (sma), which does infact obscure the important information of the position of the maximum height of the peak.

Additionally, we would like to note that all the raw crosslink data is provided in an accessible processed format in our ArrayExpress entry, for those who wish to perform their own analyses.

2.8 Author's response, Figure 1 Comparison of different window size for Gaussian smooth and simple moving average for Prp16 and Prp22. Note that the scale here is arbitrary because after smoothing with different windows the height of the lines will differ, here we just show them on the same scale for easier comparison. Raw crosslinks for the data are shown below:

Author action:

At the first instance, where we use the Gaussian smoothing, we have added a short explanation into the revised manuscript: "... While nucleotide-resolution histograms convey precise positional information, as seen in the case of SmB data, it can't be used to present data across multiple experimental conditions on a single plot since overlapping bars would not be visible. Therefore, we present the data from helicase psiCLIP experiments as Gaussian smoothed curves with a window size

of 10nt, which reflects well the crosslinking trajectory (Figure 3b), and is in agreement with the 9nt footprint of the two RecA domains observed in the structures of DEAH-box helicases...". In addition, we have included an explanation into the 'materials and methods'.

We have included an extra panel in Figure 3, where we first present the helicase psiCLIP, to show the reader the effect of smoothing on the raw data.

This is most apparent in Figure 4i, where there are more -FLAG than +FLAG crosslinks upstream of the branchpoint (stem-loop panel). This clear difference is unfortunately not visible in Figure 4h, where data are normalised and Gaussian-smoothed.

2.9 Author's response:

In this particular instance it is true that the upstream crosslinking is detected more in -FLAG condition. As described in the text, we observed crosslinking about 30nt upstream of the brA, which was regarded as background signal because we see it prominently in -FLAG samples (Figure S4). A similar behaviour of accumulation of truncation events can be seen in the -FLAG condition for the +40nt extension (without stem-loop). **This is a rare case where this matters for the interpretation of the data and is exactly why the raw crosslinks are presented here in a histogram format.**

Author action:

We amended figure 4i to show +FLAG and -FLAG experiments in separate tracks and show the full region of the stem loop.

Second, the use of different Y axis scales is simply misleading. In Figure 4, for example, when one compares panels e and f, it is clear that the difference is the peak at ~30, which is present only in the WT; otherwise the profiles of the WT and 20nt extension are almost identical.

This is readily apparent when a horizontal line is drawn at $y=0.025$. The same holds for panels g and h, with only a dip in numbers before the stem-loop in panel h. A plot of the difference would be more revealing.

2.10 Author's response:

The reviewer raises a valid concern, however we would like to refer to our response 2.4 which highlights the difference in crosslinking efficiency and describes that the crosslinking profile is not generated by a specific crosslinking efficiency of a particular nucleotide at maximal UV dose, but that the dose was titrated in order to ensure we can identify the crosslinks across all binding positions, without much bias for any particular region of the binding sites. The iCLIP experiments inevitably differ in their cDNA complexity, an aspect that is inherent to all sequencing methods, but particularly pronounced in CLIP due to the many steps that can lead to technical variation - and this needs to be normalised to enable comparisons. In most analyses in our study, the positional profile is of interest, rather than absolute amount of crosslinking - in such a case, it is a standard that separate CLIP experiments (for example when using a different splicing substrate here) are "normalised" to their highest peak for visual comparisons, as shown in Figure 4. For other examples, see Fig 7, Feng et al., 2019, DOI: 10.1016/j.molcel.2019.02.002.

Additionally, we have included a supplementary figure that shows all four extensions on the same scale to emphasise that binding is indeed expanded towards the downstream region, which is still intronic in the extended transcripts, and sharply drops in the exon 2 of all transcripts.

Author action:

We have included the above figure as an additional panel in Figure S5e, thus showing all mentioned experiments on the same scale.

Figure 3: The transient binding of Prp16 to the C complex was investigated by Prp16-psiCLIP on three different C complex preparations. However, the authors make no mention of any attempt to determine the purity of these complexes. This would be important, in particular as the AC substrate is expected to generate significant amounts of C* complex.

2.11 Author's response:

We appreciate this concern which is a general misconception in the splicing field that was clarified by the cryo-EM structures. The studies that have previously shown evidence for formation of C* complexes on 3'SS-UAC substrates have generally failed to perform factor-specific purifications but rather used just the substrate or spliceosome components present in all complexes. When factor-specific purification is performed, even with exon ligation factors, the 3'SS-UAC substrate produces complexes that by cryo-EM are entirely in the C conformation. Indeed cryo-EM suggests that at steady state there are no significant levels of stable C* complex formed on the AC substrate (Galej et al. 2016, DOI: 10.1038/nature19316; Wilkinson et al., 2020 and references therein DOI: 10.1146/annurev-biochem-091719-064225). Additionally, Prp16 would not be expected to associate with C* complexes as these have Prp22 bound at the same peripheral position after release of Prp16 that was binding there before (protein location, not RNA location). Unfortunately, with the setup of our method that requires denaturation to access the protein of interest in the FLAG-pulldown, we are unable to capture intact spliceosomes of the state that we are investigating to probe them further.

Nevertheless, we have done some pulldown experiments on Prp16 using spliceosomes assembled on *UBC4-AC* without crosslinking under less stringent conditions, which shows that Prp16 step specifically enriches for step-one spliceosomes (lariat-intermediate) but does not pull-down any early stage spliceosomes (pre-mRNA).

Author action:

We have included the pull-down experiment into Figure S4 as panel a and added some clarifications in the main text, too.

Figure 4: Prp16-psiCLIP was used to answer explicitly the question of how Prp16 acts from a distance downstream of the 3' splice site to pull the U2 branched structure out of the catalytic centre.

However, the experiment fails to capture time-resolved dynamics of Prp16 functions. These experiments cannot distinguish between simple binding/dissociation/rebinding events and the possibility that Prp16 moves in a 3'-to-5' direction on the pre-mRNA – a major question that the authors had initially set out to answer.

Thus, compared with previous biochemical and structural data, we do not learn anything new about the mechanism of action of Prp16.

2.12 Author's response:

We apologise that it was not absolutely clear that insights were gained from the two distinct sets of experiments. First, our extension experiments (Figures 4a,b,c and e,f,g) were performed, as we state "... To test the constraints on Prp16 binding ...". As we ourselves state "... it remains unclear if the broad binding reflects movement during translocation or other forms of dynamic contacts, such as binding, release and re-binding ...". Therefore we feel we have described our data accurately. We also describe a new finding about the mechanism of action of Prp16 "... Taken together, these results indicate that Prp16 initially binds downstream of the brA and contacts the substrate wherever it is accessible as a single strand. Most likely, the longer the sequence between the brA and the 3' SS, the more accessible RNA is available ...". This and the fact that the same profile is obtained for wild-type and dominant negative helicases in turn, is inconsistent with a simple processive translocation model for the activity of these helicases. We suggest a stochastic initial binding on any available single stranded RNA and have now emphasised this more within the discussion. Previous experiments could not make such a conclusion because they looked only at one RNA substrate.

Next we use Figure 4d,h, our stem-loop experiment, to explore the question of whether Prp16 acts from a distance. We base our conclusions on this experiment, however, we do not claim to follow the helicase in real time to definitively answer the question, as this is a very difficult one, but our experiment adds further weight to the hypothesis.

Author action:

We adjusted the discussion to emphasise on the stochastic binding model for both Prp16 "... Additional single-molecule and chase experiments are necessary to determine if the broad binding we observed reflects movement during translocation or other forms of dynamic contacts, such as binding, release and re-binding. However, since broad contacts were detected for both the WT helicase and the ATPase mutant, which is likely impaired in translocation, (Figure 3e,f and Figure 4e-g) we propose that the broad binding we observe reflects, at least in part, multiple rounds of binding and dissociation by one or more helicase molecule [...] The 5' arm of the stem loop showed very little crosslinking and the 3' arm showed reduced crosslinking, suggesting that the stem-loop sequence indeed formed a secondary structure that hindered Prp16-binding. Since RBPs preferentially crosslink to single-strand RNA41, the small amount of crosslinking observed towards the 3' end of the 3' arm (Figure 4i) suggests that the Prp16 helicase mutant, which retains residual ATPase activity, may have partially unwound the stem loop. Even so, the impaired Prp16 binding we observe is consistent with a model in which Prp16 remodels the branch helix without translocating fully through the brA ..."

Nevertheless, the authors make the interesting observation that Prp16 shows comparatively few crosslinks to exon 2 sequences, irrespective of the variation in length of the intron between the branchsite and the 3' splice site. Even so, the authors do not discuss the reason for this apparent discrimination against exon 2 sequences. This would have been desirable, as it stands in complete contrast to the behaviour of Prp22.

2.13 Author's response:

We thank the reviewer for this important point that was also raised by reviewer 1 in a slightly different context and would like to refer to **response and action 1.3** according to which we clarified intronic (and exonic) binding for Prp16. It is true that Prp22 exhibits more exon 2 crosslinking which is necessary for Prp22 to bind as it will release the spliced mRNA from this position. For Prp16 instead, the main binding site is within the 45nt downstream of the brA that are the common distance for a yeast unstructured intron.

Author action:

In addition to the action of 1.3, we made the discussion about Prp22 a bit clearer to point out the exonic binding.

In Figure 6, the peak of Prp22-K512A in panel c, marked with an asterisk, has an equivalent peak in panels a and b, at a similar position and with a similar intensity (related to the Y-axis scaling problem; see above, "The data analysis..."). However, in the text it is stated that the peak of Prp22-K512A in the second exon is enriched in the canonical substrate (panel c) as compared with the dG substrate (panel b).

2.13 Author's response:

This is well noted and we have now adjusted the figure and the text accordingly.

Author action:

Both Figure 6 and text have been adjusted.

It is difficult to follow the authors' argument in the discussion (p. 20) when they say that their psiCLIP data on Prp22 and Prp18 depletion support the proofreading mechanisms proposed before. To establish a causal relationship would require demonstration that the dG substrate is in fact an erroneous substrate that is eliminated by proof reading.

2.14 Author's response:

We apologize that we find a confusion of terminology here. In the classical sense, proofreading is used in the context of spliceosomal helicases proofreading the splicing substrate by not promoting progression of erroneous substrates e.g. Prp22 proofreading the 3'SS before exon ligation (Mayas et al. 2006, DOI: 10.1038/nsmb1093). Reviewer 2 is correct that this could in principle be the case for the dG substrate. However, here we propose to expand the term proofreading to include also the protein composition of the spliceosome - that the spliceosome is properly assembled and ready to continue with the splicing reaction. We realise this is confusing, and have now changed our wording in the paragraph.

Please also refer to our **response 3.5** to reviewer 3 who suggests an alternative explanation of our finding.

Author action:

We have adjusted the text in the discussion to make it clearer "... Similar to Prp22's proofreading activity for correct 3'SS selection, wild-type Prp22 may not bind stably in the absence of Prp18 and therefore prevents potential erroneous exon ligation by incompletely assembled spliceosomes ..."

Moreover, if I understand the experiments correctly, the authors' conclusions about apparent proofreading effects was based simply on mRNA reads, and had nothing to do with their psiCLIP approach. Thus, they simply confirm what has already been shown by others.

2.15 Author's response:

As described in **response 2.14** we describe Prp22 dependent rejection of spliceosomes lacking the second step factor Prp18 which has not shown before in a ATPase dependent way. One of our key results is that the ATPase deficient Prp22 mutant is able to proceed with splicing in the absence of Prp18. Herefore, we use reads from psiCLIP showing that splicing is happening at the canonical site. The additional information psiCLIP provides is that the binding profile is not changed when using the ATPase deficient mutant but resembles exactly the profile obtained in the C* complex. This comparison could not have been done with any other method. We think that this insight is valuable for future research on Prp22 proofreading and how the helicase is affected by auxiliary factors, and **in fact we were already contacted by a researcher from Jon Staley's lab at the University of Chicago, who was very interested in our Prp18 results on the basis of our pre-print and is already following up our experiments.**

Reviewer 2 Minor concerns:

Author action:

We have also fixed all the issues listed below:

One of the authors (SMF, p. 1) is actually thanked in the Acknowledgements for reagents (p. 21).

Textual: some sentences are difficult to comprehend. For example, p. 4, 1st paragraph of Results. While "conventional iCLIP" is explained, it is not clear what "the method" is that was adapted. On p. 14, bottom, as a conclusion, the authors state "[..] that psiCLIP has a large dynamic range to test hypotheses about helicase remodelling [..]". There is little meaning to these words: what is the large dynamic range and how is it related to hypotheses-testing? On p. 16: the "Cryo-EM structure of [..] shows the helicase in green [..]". On p. 3, last paragraph: what do the authors mean by "global rules of splicing"? While this is a catchy phrase, it is meaningless without a precise description.

We agree that the term 'dynamic range' was not needed in this context, and we have adjusted the text now. We used the phrase 'global rules' when referring to the published transcriptome-wide methods but have edited the text now for clarifications.

It is essential to quote the PDB ids of the structures shown in the figures (Figures 2h, 3e, and 5a,b). -
We have now quoted PDB IDs.

Figure 1e: the boxes on the blots are counterproductive, as they introduce a visual bias for the reader. What are the numbers associated with the quantification?

We find the box needed to show the membrane area selected for cDNA library preparation from the selected condition for Prp22 crosslinking. If the reviewer could suggest an alternative way to show this

area, we'll be happy to change. The numbers show the doses of UV (in 100 x $\mu\text{J}/\text{cm}^2$), as explained in the legend.

On p. 7 it should be mentioned that the Sm proteins have also been modelled in crystal structures (not cryo-EM; that is, at least the listed references describe crystal structures) - We now added 'crystal and cryoEM, as we included a cryoEM study (Bai et al., 2018).

Which pre-mRNA was used in Figure 2e? - The short UBC4 substrate was used, as we now clarify in the legend.

Figure 3c: The difference between AC and AG symbols can only be guessed at. - This has been made clearer in revised Figure 3.

Figures 4 e-h: Are the data quantitatively comparable in the horizontal axis? In other words, are the total numbers of crosslinks similar in the blue areas? Looking at Figure 4i, I would presume not. However, the authors do not address this point. An analogous point can be made for Figures 3c and d. Therefore I consider that a thorough re analysis of the data is essential. - We hope that we have addressed concerns about Figure 4 in Author's response 2.10, and that we have addressed concerns about Figure 3 in Author's response 1.2 and 2.7.

Figure 4i: The zero on the X-axis should be a ten. - Figure 4i has been amended with all suggestions from all reviewers.

Figure 5d: Incomplete labels (cf 3a). - All C* complex reactions were necessarily conducted with dG splicing substrate to ensure the stalling of the spliceosome in this state, and all P complex reactions were performed with a WT substrate to enable the spliceosome to proceed past exon ligation. Therefore in the Figure 5 graphs we chose to label the purified complex, as opposed to 3'SS used. To clarify we have added the 3'SS to the title of each graph in Figure 5c.

p. 15, bottom: The number 0.0003% looks strange, as it suggests an accuracy that I do not believe can be achieved experimentally. How many reads lie behind this number? - This number refers to data shown in graphical format in Figure S5b. You are correct, this is a proportion not a percentage. We have now corrected the number in the text to 0.03% (for reference this figure comes from 46 junction reads/133,468 total reads).

p. 16, line 3: How was the RNA "predicted"? We have now clarified in the figure legend (Figure 3) that the dotted line gives simply an indication where the RNA could be located and shows the connection rather than being a proper prediction of the RNA path. The publication containing the cryoEM structure of complex C (Galej et al., 2017, DOI: 10.1038/nature19316) gives the possible number of unstructured nucleotides that could be located in that region but does not state how this number was predicted.

p. 20, centre: What would constitute an "incompletely assembled spliceosome"? - See response and actions **2.14**

p. 21: "N.M.L" - Corrected

p. 26, bottom: Tarasov must be cited for Sambamba, and not for the strandedness of psiCLIP. Also, further down, what were the reads mapping to, if not to the custom transcriptome? The idea of the "internal spike" is interesting, but more details need to be supplied. - We have added further information now in our first section of results:

"... Notably, we found that between psiCLIP experiments for the same protein, the proportion of endogenous yeast RNA originating from the yeast extract remained similar, even when binding to our

substrate sequences did not (data not shown). This allowed us to use the proportion of yeast genome mapping RNA in the library as internal normalisation, much like a “spike in”...”

Reviewer #3:

Pre-mRNA splicing is an extremely complex and highly dynamic process that takes place in a multi-megadalton RNA-protein complex, the spliceosome. The spliceosome assembles in a highly ordered series of steps that involve the addition of many factors, structural rearrangements, two catalytic reactions, and finally release of the spliced RNA and dissociation of the complex. This manuscript describes the development and application of a sophisticated biochemical approach referred to as psiCLIP to pinpoint the position within RNAs that proteins bind at different stages in the splicing cycle. This technically challenging approach involves assembling spliceosomes in vitro, UV irradiation to covalently crosslink proteins to the RNAs in the complex, recovery of the crosslinked protein-RNA adducts, followed by cDNA library preparation, high-throughput sequencing, and bioinformatic analysis of the data. Although a similar approach has been used to analyse protein-RNA interaction in live cells, the protocol had to be adapted for this novel in vitro application. This involved trials to titrate the UV treatment, RNA digestion and various reagents, and included meticulous use of controls.

Eight RNA-stimulated ATPases are implicated in promoting structural changes and as fidelity factors for quality control of the process. The work here focuses on two of these, Prp16 and Prp22. Following validation of the procedure using the well characterised SmB protein, interactions of the Prp16 protein prior to the second catalytic reaction were mapped by stalling spliceosomes at complex C, using a mutant form of Prp16 and with two different pre-mRNAs with canonical or mutated 3' splice sites. Crosslinks for both wild-type and mutant Prp16 were mapped throughout the region downstream of the intron branch site as far as the 3' splice site, even when the length of this region was extended by 40 nucleotides. Moreover, the finding that a stem-loop structure inserted in this region did not prevent splicing is thought to indicate that Prp16 does not need to translocate through this region.

Analogous experiments were performed to map Prp22 interactions in complex C*, immediately prior to exon ligation, and in complex P immediately after the second trans-esterification reaction. Prp22 appears to bind a wide region from the branchpoint downstream into the second exon in complex C* but with more localised binding in the second exon in complex P. Most interestingly, depletion of the second step factor, Prp18, resulted in loss of WT Prp22 binding but not of the mutant Prp22 binding. This is interpreted to indicate that binding by WT Prp22 is destabilised upon Prp18 depletion, which is compatible with structural studies.

In summary, this is a sophisticated biochemical analysis, with interesting results that confirm, extend and clarify previous observations and which will be valuable to the splicing community. From a methodological point of view, the ability to compare protein-RNA interactions in successive splicing complexes offers a more dynamic view of these interactions than was previously possible and it would be informative to extend this approach to other splicing factors. The work seems to have been performed to a very high standard and, for the most part, the authors have carefully considered alternative interpretations of their data and appropriately qualified their conclusions. I suggest below some **minor considerations**:

3.1 Author's response:

We would like to thank reviewer 3 for their close attention to our manuscript and the positive feedback regarding the rigour of experiments. We are grateful to receive suggestions for an alternative interpretation of our results, and think that these have improved our manuscript substantially.

Reviewer 3 minor concerns:

1) as Prp22 was observed to interact with the second exon just downstream of the 3' splice site, it would seem appropriate to discuss how this relates to binding of other proteins, e.g. Prp8 and Prp18, in the same region.

3.2 Author's response:

We agree that the additional requested information would add more value to the finding and have included interactions with Prp8 into the manuscript now. Unfortunately, we couldn't find evidence in the literature for where Prp18 would bind precisely. There is evidence of a genetic interaction with both splice sites (Crotti et al., 2007, doi: 10.1101/gad.1538207), but in published structures (Fica et al., 2017, doi: 10.1038/nature21078) we could only find a small loop of Prp18 coming close to the active site and therefore to the 3'SS.

Author action:

We have incorporated a short section describing the binding positions of Prp8 in relation to Prp22 into the results section of Prp22.

2) On p19/20 in the Discussion, binding of Prp16 or Prp22 over extensive regions of RNA is interpreted as reflecting dynamic binding and dissociation or translocation of the proteins along the RNA molecules. As the data come from populations of splicing complexes, widespread cross-links could reflect heterogenous binding among distinct complexes in the population rather than movement within individual complexes, although it might indeed also reflect snapshots of translocating or repeatedly binding Prp22 on different pre-mRNAs. This should be clarified.

3.3 Author's response:

We acknowledge that there is a possibility to have captured distinct complexes in the population and have highlighted this in the manuscript accordingly. In the case of Prp22, we must stress that whilst Prp22 may be binding heterogeneously within C* and P complex preparations respectively, comparisons between the two populations are very important to our argument against a simple directional processive translocation being the main mechanism of Prp22 action. This is because the intronic binding seen in C* complex is 5' to the exonic binding seen in P complex, and *in vitro* work has shown Prp22 incapable of 5' -> 3' translocation. Furthermore, that similar binding patterns are seen for both Prp16 and Prp22 ATPase-deficient, and therefore helicase activity-deficient, mutants, would suggest that the binding we see - whether in a heterogeneous population or not, is not due to translocations.

Author action:

We discuss the likelihood of this hypothesis in the discussion section now.

“... The high reproducibility of binding profiles observed at specific substrate positions in several replicates and across stalling conditions (Figure S4b-g) likely represents an average of interactions in defined complexes, rather than resulting from heterogeneous binding or processive translocation events across multiple complexes. Thus we interpret the broad binding as evidence that the helicases can bind at multiple defined regions along the substrate. In the C* complex we detected Prp22 binding on the intron before the 3' SS and on the second exon downstream of the 3'SS . The strong signal

observed for the wild-type Prp22 protein as well as the broad profile seen for the ATPase-deficient K512A mutant, which is unable to translocate, suggest multiple rounds of stochastic binding and dissociation, rather than a single ATP-dependent translocation and dissociation event...”

3) bottom of page 19, there is a confusing sentence: “In complex P, Prp22 binding shifts to a narrower region in 3’ exon, and this exonic binding pattern is similar between complex C* and complex P.”

3.4 Author’s response:

This was also pointed out by reviewer 1 and is addressed in **response and action 1.4**.

4) Fig 6 and on p20, “Prp18 depletion abrogates WT Prp22 binding...” This interesting result might be explained by an alternative (hypothetical) role for Prp18, modulating the ATPase activity of Prp22. In this scenario, the absence of Prp18 might cause the ATPase of Prp22 to be over-active, resulting in rejection of spliceosomes at that stage, whereas the mutant Prp22 with reduced ATPase activity bypasses this proof-reading step and bypasses the need for Prp18. Has the effect of Prp18 on Prp22’s ATPase activity been tested?

3.5 Author’s response:

We thank reviewer 3 for this additional possible explanation and have included the additional hypothetical role of Prp18 into the manuscript. Unfortunately, we could not find any literature that investigated the effect of Prp18 onto Prp22’s ATPase activity. As Prp18 and Prp22 do not directly interact, the activity test would have needed to be performed on a fully assembled spliceosome in a setting that can distinguish between ATPase activity of Prp22 and other helicases such as Brr2 and Prp43. To date we are not aware of how the ATPase activity could therefore be tested.

Author action:

We included an additional short section about the additional role into our revised discussion section. “... Depletion of Prp18 may indirectly increase the ATPase activity of Prp22, promoting dissociation and reducing stable binding to the substrate ...”

5) A minor point: in Fig 5d the results for the C, C* and P complexes are plotted in the opposite order to the legend and to the order of splicing complex progression, which could be confusing.

3.6 Author’s response:

We acknowledge that this is confusing and have changed it accordingly.

Author action:

The legend order and bar order now match and follow the order of splicing progression.

REVIEWERS' COMMENTS

Reviewer #1 (Remarks to the Author):

Reviewer #1 comments on revisions by Strittmatter and Capitanchik et al.:

The authors have answered and addressed all concerns I previously raised in my review of the manuscript in their extensive response letter.

In addition, the authors have also taken any criticism, where appropriately raised by the reviewers, and re-assessed their interpretation of the data and adapted the manuscript where necessary by rewriting text passages and changing figure composition or data presentation to better illustrate their findings .

In my opinion the review has improved the manuscript substantially. As already stated by Reviewer #3 and me, this study not only confirms and clarifies previous observations but also extends them. The changes wrought by the authors on the manuscript should also better illustrate the extension of the previous understanding, which has been less apparent for Reviewer #2.

Reviewer #2 (Remarks to the Author):

The manuscript by Strittmatter and Capitanchik et al. was thoroughly revised by the authors. I can only congratulate them on their careful scholarly work. Especially, in that every point raised by me and the other reviewers was treated exhaustively and objectively, and incorporated where required in the revised manuscript. It was clear from the first submission that a partial rewrite was necessary. This now resulted in a manuscript that is clear and streamlined to the major findings. The authors also did not shy away from incorporating ideas suggested by the reviewers. The figures improved a lot, in particular the cartoons of structures. The manuscript will now be accessible to a large audience.

I take it for granted, that panels e and f in Figure S7 were inadvertently reversed in the first draft.

There are two minor points that I leave to the discretion of the authors to take care of. First, in Figures 5a/b, 6a and 7 as well as Supplementary Figure S8, Prp22 is coded in different colours, apparently with the aim to differentiate the states. This introduces a new level of coding complexity that is remotely related to a heatmap. I think it is unnecessary and, minimally, should be described in the legends. Second, in some of the equilibria shown in Figures 6a, 7, and S8, the cartoons of the factors coming in and/or out are missing.

Reviewer #3 (Remarks to the Author):

The revised manuscript is significantly improved. In particular, the methodology and conclusions are more clearly stated and my concerns have been addressed. I have two comments:

Page 2: This new text is unclear and potentially confusing "the spliceosome catalyzes attack of the branchpoint adenosine (brA) at the 5' splice site (5'SS)". The spliceosome catalyzes attack "BY" the brA at the phosphodiester bond at the 5'SS.

Regarding my point 4 (authors' response 3.5): The text inserted in response to this, on page 12 is a bit confusing: "Depletion of Prp18 may indirectly increase the ATPase activity of Prp22, promoting dissociation and reducing stable binding to the substrate. Alternatively, in the absence of

Prp18, Prp22 may dissociate from the spliceosome and may not rebind." The following would make more sense: In the absence of Prp18, Prp22 may dissociate from the spliceosome and may not rebind. Potentially, Prp18 may indirectly increase the ATPase activity of Prp22, promoting its dissociation and reducing stable binding to the substrate. Indeed, structural studies have shown that the absence of Prp18 destabilizes the C* and P conformation....

Reviewer #4 (Remarks to the Author):

The authors apply iCLIP to purified spliceosomes in order to better characterize the interactions of RNA helicases that are poorly defined in other structural analyses. The CLIP analyses have been well performed and analyzed. Crosslinking approaches generally give more "fuzzy" results than crystallography, in which homogeneity is enforced by the lattice, or cryoEM, in which class averages are generally based on highly-selected subsets of images. In contrast, here we see the range of interactions present within the particles. This inevitably makes the data more open to interpretation, but it provides valuable information on the functional interactions of factors that are too labile or mobile for other approaches.

The authors appear to have done a good job in responding to the extensive initial round of comments and I am happy to support publication.

Minor point:

- 1) There are minor formatting problems with line spacing and in-text reference numbers.

“PsiCLIP reveals dynamic RNA binding by DEAH-box helicases before and after exon ligation” Response to Reviewers

We would like to thank all of the reviewers for their thoughtful feedback on our work, which helped us to improve the paper substantially from our first submission. We are happy to see that we have met the expectations of all reviewers with our first resubmission and that reviewer 4 appreciates our contribution from a CLIP perspective. Where the reviewers have made additional comments on formatting and a few text edits, please find our responses below in black.

Reviewer #2 (Remarks to the Author):

...I take it for granted, that panels e and f in Figure S7 were inadvertently reversed in the first draft...

This is correct, and Figure S7 e and f are now correctly placed.

...There are two minor points that I leave to the discretion of the authors to take care of. First, in Figures 5a/b, 6a and 7 as well as Supplementary Figure S8, Prp22 is coded in different colours, apparently with the aim to differentiate the states. This introduces a new level of coding complexity that is remotely related to a heatmap. I think it is unnecessary and, minimally, should be described in the legends...

We agree with the reviewer that the differential colouring of Prp22 was misleading in our original draft. We have changed all Prp22-related colouring to shades of purple to avoid confusing the reader.

...Second, in some of the equilibria shown in Figures 6a, 7, and S8, the cartoons of the factors coming in and/or out are missing...

The equilibria shown in Figure 6A represent only the main conformational states and are only meant to illustrate composition of different complexes. Since the binding pathway for Prp18 and Slu7 is complex (see Wilkinson et al., bioRxiv 2020) we do not believe it is helpful, or even possible, to clearly illustrate factors coming in and out in Figure 6A. Moreover, we believe such illustration is not necessary and would distract from the main point of the figure. We have, however, added binding of incoming Prp22 in Figure 7 and of Prp22 and Prp18 in Supplementary Figure 8, as the reviewer suggested. We agree that in these figures it is helpful to illustrate incoming factors.

Reviewer #3 (Remarks to the Author):

...Page 2: This new text is unclear and potentially confusing “the spliceosome catalyzes attack of the branchpoint adenosine (brA) at the 5' splice site (5'SS)”. The spliceosome catalyzes attack "BY" the brA at the phosphodiester bond at the 5'SS...

...Regarding my point 4 (authors' response 3.5): The text inserted in response to this, on page 12 is a bit confusing: "Depletion of Prp18 may indirectly increase the ATPase activity of Prp22, promoting dissociation and reducing stable binding to the substrate. Alternatively, in the absence of Prp18, Prp22 may dissociate from the spliceosome and may not rebind." The following would make more sense: In the absence of Prp18, Prp22 may dissociate from the spliceosome and may not rebind. Potentially, Prp18 may indirectly increase the ATPase activity of Prp22, promoting its dissociation and reducing stable binding to the substrate. Indeed, structural studies have shown that the absence of Prp18 destabilizes the C* and P conformation....

We thank the reviewer for the suggestion to clarify the above statements. We have implemented both in the final draft.

Reviewer #4 (Remarks to the Author):

1) There are minor formatting problems with line spacing and in-text reference numbers.

We agree with the reviewer, and have improved formatting and in-text referencing in the final draft.